# Spatial-temporal trends and risk factors for undernutrition and obesity among children (<5 years) in South Africa, 2008–2017: findings from a nationally representative longitudinal panel survey

Benn Sartorius [1,2,3] Kurt Sartorius,[2] Rosemary Green,[4] Elizabeth Lutge,[5] Pauline Scheelbeek [4] Frank Tanser,[2,6,7] Alan D Dangour,[4] Rob Slotow[8]

For numbered affiliations see end of article.

**Correspondence to**
Dr Benn Sartorius;
benn.sartorius1@lshtm.ac.uk

## ABSTRACT

**Objectives** To assess space-time trends in malnutrition and associated risk factors among children (<5 years) in South Africa.

**Design** Multiround national panel survey using multistage random sampling.

**Setting** National, community based.

**Participants** Community-based sample of children and adults. Sample size: 3254 children in wave 1 (2008) to 4710 children in wave 5 (2017).

**Primary outcomes** Stunting, wasting/thinness and obesity among children (<5). Classification was based on anthropometric (height and weight) z-scores using WHO growth standards.

**Results** Between 2008 and 2017, a larger decline nationally in stunting among children (<5) was observed from 11.0% to 7.6% (p=0.007), compared with thinness/wasting (5.2% to 3.8%, p=0.131) and obesity (14.5% to 12.9%, p=0.312). A geographic nutritional gradient was observed with obesity more pronounced in the east of the country and thinness/wasting more pronounced in the west. Approximately 73% of districts had an estimated wasting prevalence below the 2025 target threshold of 5% in 2017 while 83% and 88% of districts achieved the necessary relative reduction in stunting and no increase in obesity respectively from 2012 to 2017 in line with 2025 targets. African ethnicity, male gender, low birth weight, lower socioeconomic and maternal/paternal education status and rural residence were significantly associated with stunting. Children in lower income and food-insecure households with young malnourished mothers were significantly more likely to be thin/wasted while African children, with higher birth weights, living in lower income households in KwaZulu-Natal and Eastern Cape were significantly more likely to be obese.

**Conclusions** While improvements in stunting have been observed, thinness/wasting and obesity prevalence remain largely unchanged. The geographic and sociodemographic heterogeneity in childhood malnutrition has implications for equitable attainment of global nutritional targets for 2025, with many districts having dual epidemics of undernutrition and overnutrition. Effective subnational-level public health planning and tailored interventions are required to address this challenge.

## Strengths and limitations of this study

► Uses data from a nationally representative repeated panel data at individual/household level over a 10-year period (five survey waves).

► Employed a fully Bayesian space-time shared component model to produce more stable estimates of malnutrition burden at provincial and district levels among children under 5 years of age in South Africa.

► Panel design allows assessment of change in malnutrition burden within the same individuals/households observed at multiple time points.

► Missing or invalid weight/height measurements may have introduced selection bias if not missing at random, and may thus have affected both the internal validity and the representativeness of the findings.

► As primary panel study was not designed/powered for provincial and lower geographic-level analysis, we cannot discount the resultant impact on precision/random variability when analysing at provincial/district level (administrative tier just below province) and further stratification by sociodemographic correlates.

## BACKGROUND

Despite reductions in malnutrition 150.8 million children (22.2%) under 5 are stunted and a further 50.5 million children are wasted.[1] Furthermore, rapidly rising trend in overweight and obesity in children and adults[2–5] has emerged as one of the most serious global public health issues of the 21st century.[6] Sub-Saharan Africa (SSA) has among the highest levels of child malnutrition (Child malnutrition is defined as a pathological state as a result of inadequate nutrition, including undernutrition due to insufficient intake of dietary energy and other key nutrients resulting in stunting (low height for age (HA)) or wasting (low weight for length) and overweight and obesity due

to excessive consumption of dietary energy and reduced levels of physical activity) globally. This problem is particularly illustrated by South Africa,[7] a middle-income country with high levels of wealth/economic inequality that is undergoing rapid socioeconomic and lifestyle changes that have precipitated a nutritional transition, high prevalence of overweight/obesity in children.[8] The dual burdens of undernutrition and overweight/obesity are not distributed in a spatially homogenous manner,[9] and the health risks associated with malnutrition vary by age, gender, ethnicity and geographical location.[10]

Progress to tackle all forms of child malnutrition remains much too slow.[1] In order to support the delivery of public health interventions that will be most effective at reducing malnutrition, an understanding of the geographical distribution of malnutrition is required. Limited data are collected at lower administrative unit level making it difficult to identify specific groups of high-risk individuals, and thus determine the most suitable and cost-effective opportunities and solutions. Previous studies of nutritional status of the South African population have mostly focused on adults.[11] [12] Here we use a large, nationally representative data from multiple rounds of the National Income Dynamics Study (NIDS) over the period 2008–2017 to assess space-time trends in the burden of malnutrition and associated risk factors among children under 5 years of age in South Africa.

## METHODS
We include a Strengthening the Reporting of Observational studies in Epidemiology statement[13] checklist in online supplementary material 1.

### Data
Data were taken from the five panel (cross-sectional) waves of the South African National Income Dynamics Study (SA-NIDS)[14] [15] (http://www.nids.uct.ac.za/nids-data/data-access; https://www.datafirst.uct.ac.za/data-portal/index.php/catalog/NIDS/), the first national panel study in South Africa. SA-NIDS was undertaken by the South African Labour and Development Research Unit based at the School of Economics at the University of Cape Town. The surveys took place in 2008, 2010–2011, 2012, 2014–2015 and 2017. These are named waves 1–5 respectively. A detailed description of the data collection methods can be found elsewhere.[14] In short, a stratified, two-stage random cluster sample design was employed to sample households for inclusion at baseline using proportionally allocated stratification, based on the 52 district councils (DC) in South Africa.[14] Within each DC (primary sampling unit), clusters of dwelling units were systematically drawn. The household-level response rate was 69% and the individual response rate within households was 93%. Survey enumerators attempted to collect weight and height measurements of all individuals (including children) in selected households.

### Study population
We restricted our analysis to children <5 years of age.

### Outcomes
We calculated HA and body mass index (BMI)-for-age (BA) z-scores using the WHO 2007 growth standards.[16] [17] We generated z-scores by transformation of child anthropometric data using the 'lambda mu sigma' method ('zanthro' function in Stata V.15). As recommended, weight for length was used in children 0 to <2 years of age, and BA in children 2 years of age and older.[18] We defined obesity as weight-for-length z-score ≥+2 for children under 2 years of age and BA z-score of >2+ for children aged 2 and older.[18] We defined wasting as weight-for-length z-score <−2 for children under 2 years of age and thinness as BA z-score <−2 for children 2 years and older. Stunting was defined as HA z-score of <−2.

### Geographic and sociodemographic variables
To identify relevant inequalities, undernutrition and obesity indicators were stratified temporally (survey year), geographically (province and residence location type: urban informal settlements, urban formal, tribal/rural) and by important sociodemographic categories (gender: female/male; ethnicity: Black/African, coloured, Indian/Asian, White/Caucasian; maternal: age; education status; BMI; household socioeconomic status (SES) (income) classified into quantiles (1=lowest, 5=highest)).

### Data analysis
Analyses were performed using Stata software V.15 (StataCorp. 2017. Stata Statistical Software: Release 15. College Station, TX: StataCorp). Given the multistage random sampling design of the primary study, clustering and survey design effects were accounted for using sample weights to estimate SE and 95% CIs around mean anthropometric z-score point estimates, both overall and stratified by other sociodemographic variables such as ethnicity and gender, SES and residence location type. Extrapolated population totals of malnourished children (<5) by yearly age were estimated using the survey weights.

### Space-time Bayesian modelling
We assessed for the presence of univariate and bivariate spatial autocorrelations for the three anthropometric classifications using Moran's I statistics. This analysis was performed using GeoDa.[19] Based on these tests it appeared that there was no prominent bivariate spatial autocorrelation between the three measures but that each measure was significantly heterogeneous across space, warranting the use of a separate spatial-temporal model for each nutritional outcome. These additional analyses are presented in online supplementary material 2.

We employed Bayesian spatial-temporal modelling approach in an attempt to stabilise estimates at district level given that the primary sampling design was not developed to provide point estimates at this level of geographic disaggregation and resultant zero prevalence estimates for particular districts and waves. We choose a

Bayesian spatial-temporal formulation to model each of the anthropometric outcomes independently using an autoregressive approach. We employed a Bayesian hierarchical binomial model that simultaneously attempts to estimate the stable spatial and temporal structured patterns and as well as from these stable components using an unstructured space-time interaction term.[20]

Let $Y_{1ij}$, $Y_{2ij}$ and $Y_{3ij}$ be the numbers of stunted, thin and obese children, respectively, for the ith area and jth period, i=1,…,I, j=1,…,J and $n_{ij}$ the total number of children sampled in a given area and period. We assumed that $Y_{1ij}$, $Y_{2ij}$ and $Y_{3ij}$ follow binomial distributions, that is, $Y_{1ij}$~binomial $(n_{1ij}, \varpi_{1ij})$, $Y_{2ij}$~binomial $(n_{2ij}, \varpi_{2ij})$, $Y_{3ij}$~binomial $(n_{3ij}, \varpi_{3ij})$, i=1,…,53, j=1,…,5, where $\varpi$ is the risk (prevalence) of stunting, thinness or obesity in region i in period j. We define the logit of the prevalence for a given anthropometric outcome as follows:

$$\mathrm{logit}(\pi_{1ij}) = \alpha_1 + \phi_{1i} + \gamma_{1j} + \nu_{1ij}$$

$$\mathrm{logit}(\pi_{2ij}) = \alpha_2 + \phi_{2i} + \gamma_{2j} + \nu_{2ij}$$

$$\mathrm{logit}(\pi_{3ij}) = \alpha_3 + \phi_{3i} + \gamma_{3j} + \nu_{3ij}$$

$$\nu \sim \mathrm{Normal}(, \sigma^2{}_\nu), i = 1, ..., I \text{ and } j = 1, ..., J$$

$$\phi \sim \mathrm{CAR.normal}(\sigma^2{}_\phi), \text{ for } i = 1, ... I$$

$$\gamma = (\gamma_1, \gamma_2, ..., \gamma_J) \sim \mathrm{CAR.normal}(\sigma^2{}_\gamma) \alpha \sim \mathrm{Uniform}(-\infty, +\infty),$$

where $\alpha_{1-3}$ are the overall baseline risk (intercept) for each nutritional outcome, $\varphi_{1-3}$ the spatial random effects, assume intrinsic Gaussian conditionally autoregressive distributions[21] (abbreviated above as CAR. normal), whereby the spatially correlated random effect of the ith region ($\phi_i$) is based on the sum of its weighted neighbourhood values. We used an adjacency matrix of common boundaries (neighbours) of a given region when modelling this parameter. The CAR approach can also be used to model the temporal random effects. A first-order (pre and post) random walk CAR.normal, using a period adjacency matrix, was used as prior distributions for the temporal random effects, $\gamma_{1-3}$. The heterogeneous or unstructured random effects are represented by $\nu_{1-3}$ and were included to ensure sufficient flexibility for estimates in close regions that is not captured by the spatially structured terms. We assumed uniform priors for the model intercepts to ensure model identifiability. As the CAR.normal distribution is parameterised to include a sum-to-zero constraint on the random effects, we thus included a separate intercept term, $\alpha$, in each model, which were assigned improper uniform priors (on the whole real line) using the dflat() distribution function in WinBUGS. We chose inverse gamma distributions for the variance parameters above with values of 0.5 and 0.0005 as suggested by Wakefield et al[22]:

$$\sigma^2{}_\nu, \ \sigma^2{}_\phi, \ \sigma^2{}_\gamma \sim \mathrm{Gamma}(0.5, \ 0.0005)$$

To aid the interpretation of prevalence point estimates in line with WHO 2025 nutritional targets we also estimated exceedance probabilities associated with the target thresholds for each nutritional outcome, namely: 40% reduction in stunting from 2012 to 2015, reduce and maintain wasting to <5% by 2025 and no increase in obesity by 2025.[23] We employed Richardson's criterion, in which probabilities in excess of 0.8 were deemed to be significant.[24]

Survey weighted prevalences were applied to sample size totals by district and panel to obtain a survey weighted numerator count for each outcome ($Y_{1ij}$, $Y_{2ij}$, $Y_{3ij}$ above) from the binomial distribution. The space-time models were fitted in WinBUGS using Markov chain Monte Carlo (MCMC) simulation and non-informative priors. The full WinBUGS model code is provided in the online supplementary material 3. A summary of the space-time random effect posteriors is presented in online supplementary material 4. Sensitivity of the estimates to prior specification was assessed by repeating the analysis with different hyperparameters (online supplementary material 4). We used two-chain MCMC simulation for parameter estimation, a burn-in of 10 000 iterations and Gelman-Rubin statistics/plots[25] were used to assess model convergence/stability and where the Monte Carlo error for each parameter of interest was less than 5% of the sample SD (online supplementary material 5). For model validation, we first compared the observed and fitted prevalence values to assess overall model adequacy and fit (using model deviance information criterion and comparison of observed vs fitted prevalence estimate) and, second, performed an out-of-sample validation using a random 10% sample with observed data (online supplementary material 6). The model was run until the Monte Carlo error for each parameter of interest was <5% of the sample SD. Posterior prevalence estimates and 95% Bayesian credibility intervals for stunting, thinness/wasting and obesity at provincial and district levels were mapped using ArcGIS V.10.6.1 (ESRI 2011. ArcGIS Desktop: Release 10. Redlands, CA: Environmental Systems Research Institute).

### Risk factors analysis

Survey weighted two-way tabulations of key sociodemographic covariates, year and child nutritional status were performed to produce correctly weighted prevalence estimates. Tests of independence for complex survey data (weighted Pearson's $\chi^2$ test) were used to assess the significance of bivariate associations between malnutrition burden and year as well as sociodemographic covariates.

### Patient and public involvement

As this was a data analysis using secondary data from a national community-based panel survey, the development of the research question was not informed by the study subjects. Likewise, we could not involve study participants in the design of this study. Study participants were not involved in conduct of the primary study. Results will be disseminated in the form of peer-reviewed article as well as

through presentation to senior members of our National Department of Health and KwaZulu-Natal Department of Health.

## RESULTS
### Study population
The sample of children <5 years of age in the 7301 households included in the SA-NIDS survey increased from 3254 children at baseline (2008) to 4710 children in wave 5 (2017) (online supplementary material 7). With the exception of children under 1 year of age and survey wave 2 in 2010/2011, valid weight and height measurements were taken from 85% to 90% of children sampled between the ages of 1 and 5 on average (online supplementary material 7). An additional sensitivity analysis comparing distributions of various sociodemographic characteristics by missing weight/height status was also performed (online supplementary material 8). These findings suggest that children with missing weight/height were largely missing at random, with the exception of age and province. A summary of the characteristics of the study sample by year can be found in table 1.

### Temporal changes in burden of malnutrition from 2008 to 2017
Between 2008 and 2017, the prevalence of stunting among children aged under 5 years decreased from 11.0% to 7.6% (p=0.007) (table 2). Over the same period, both the prevalence of wasting/thinness and the prevalence of obesity decreased (from 5.2% to 3.8%, p=0.131 and 14.5% to 12.9%, p=0.312, respectively). The prevalence of thinness was higher (p<0.001) in children under 2 years of age (8% (95% CI 5.0% to 11.8%) in 2008; 6% (95% CI 4.1% to 9.1%) in 2017) compared with 4% (95% CI 3.2% to 6.2%) in 2008 and 3% (95% CI 2.0% to 4.5%) in 2017 among children 2 years and older . The prevalence of obesity was also higher among children under 2 years of age and increased over the study period (18.4% (95% CI 13.7% to 24.1%) in 2008 vs 21.7% (95% CI 19.3% to 24.2%) in 2017, p=0.091).

### Space-time burden of malnutrition at provincial and district levels
#### Undernutrition
In 2008, the highest prevalence of stunting was estimated in the Free State (18%), followed by Eastern Cape (14.8%) and Limpopo (14.0%). By 2017, the highest prevalence of stunting was still observed in Free State (10%), followed by Northern Cape (9.6%) and Limpopo (8.5%) (figure 1A). One district in Free State (Lejweleputswa), two in Limpopo (Capricorn; Mopani) and one each in Northern Cape (Siyanda), North-West (Dr Kenneth Kaunda), Eastern Cape (OR Tambo) and KwaZulu-Natal (Uthungulu) had a posterior median smoothed prevalence of stunting in excess of 10% in 2017 (figure 1B, online supplementary material 9). Forty-three (or 83%) of districts achieved a 17% reduction (necessary

reduction over the period to achieve 40% reduction from 2012 to 2025) in stunting prevalence from 2012 to 2017. Of these 43 districts, 19 (or 44%) significantly achieved this threshold based on exceedance probability (p>0.80).

North-West province had the highest burden of thinness/wasting in 2008 (10.1%), followed by Gauteng (9.5%) and Western Cape (8.2%) (figure 2A). By 2017, the highest burden was observed in Western Cape (at 5.8%), followed by Northern West (5.0%) and North Cape (4.9%) (figure 2B), that is, two of nine provinces were still above the 5% target threshold for wasting in 2017. There appeared to be a general gradient of higher burden of thinness/wasting in the western half of the country in 2017 (lower burden in KwaZulu-Natal and northern districts of Eastern Cape) (figure 2B). Our estimates suggest that 38/52 (or 73%) districts in 2017 were below the 5% target prevalence threshold compared with 21/52 (or 40%) in 2012. Based on exceedance probability associated with the 5% target threshold, approximately half (or 18/38) of the aforementioned districts with an estimated thinness/wasting prevalence below 5% in 2017 were below this threshold with high probability (exceedance p>0.8) (online supplementary material 9). Three of the five districts with the highest posterior median smoothed prevalence of wasting in 2017 were located in Western Cape (City of Cape Town (6.8%); Central Karoo (6.4%); Eden (6.1%)), with the remaining two in the top five located in Eastern Cape (Buffalo City (7.9%)) and Gauteng (Sedibeng (6.6%)) (online supplementary material 9).

#### Obesity
In 2008, the highest posterior median smoothed prevalence of obesity was estimated in Eastern Cape (22.5%), followed by KwaZulu-Natal (18.3%) and Western Cape (18.1%) (figure 3A). A decade later in 2017, the highest prevalence of childhood obesity was still estimated to be in the Eastern Cape (16.7%), followed by KwaZulu-Natal (15.6%) and Western Cape (15.0%). Six districts had an increase in obesity from 2012 to 2017, namely: three in Limpopo (Capricorn, Vhembe, Waterberg), one in Free State (Mangaung), one in Eastern Cape (Amathole) and one in North-West (Bojanala) (online supplementary material 9). In contrast to the wasting gradient highlighted above (higher burden in the western half of the country), the burden of obesity in 2017 appeared to be much higher in the eastern half of the country (particularly KwaZulu-Natal and Eastern Cape) (figure 3B), with the exception of certain districts in Western Cape. Eight of the top 10 highest obesity prevalence districts in 2017 were located in KwaZulu-Natal (Sisonke (21.4%), Ugu (20.8%), Uthungulu (18.6%) and iLembe (18.0%)) and Eastern Cape (Buffalo City Metropolitan (22.8%), Amathole (19.6%), Chris Hani (18.5%), OR Tambo (17.9%)). The other two districts in the 10 highest obesity prevalence districts in 2017 were located in Western Cape (Overberg (22.0%) and City of Cape Town (18.5%)) (online supplementary material 9).

**Table 1** Sociodemographic characteristics of sampled children by survey round

| Variable | Category | Wave 1: 2008 n (%) | Wave 2: 2010/2011 n (%) | Wave 3: 2012 n (%) | Wave 4: 2014/2015 n (%) | Wave 5: 2017 n (%) |
|---|---|---|---|---|---|---|
| Age (years) | <1 | 661 (20.3) | 517 (14.6) | 652 (17) | 886 (19.7) | 813 (17.3) |
| | 1–1.99 | 661 (20.3) | 621 (17.5) | 691 (18) | 875 (19.5) | 909 (19.3) |
| | 2–2.99 | 670 (20.6) | 751 (21.2) | 764 (19.9) | 863 (19.2) | 996 (21.1) |
| | 3–3.99 | 642 (19.7) | 840 (23.7) | 826 (21.5) | 914 (20.3) | 992 (21.1) |
| | 4–4.99 | 620 (19.1) | 820 (23.1) | 909 (23.7) | 960 (21.3) | 1000 (21.2) |
| Gender | Male | 1640 (50.4) | 1773 (50) | 1856 (48.3) | 2173 (48.3) | 2325 (49.4) |
| | Female | 1614 (49.6) | 1770 (49.9) | 1986 (51.7) | 2322 (51.6) | 2385 (50.6) |
| Ethnicity* | African | 2723 (83.7) | 3047 (85.9) | 3307 (86.1) | 3898 (86.7) | 4048 (85.9) |
| | Coloured | 429 (13.2) | 423 (11.9) | 455 (11.8) | 532 (11.8) | 523 (11.1) |
| | Asian/Indian | 32 (1) | 26 (0.7) | 24 (0.6) | 30 (0.7) | 0 (0) |
| | White | 70 (2.2) | 53 (1.5) | 56 (1.5) | 29 (0.6) | 0 (0) |
| Birth weight | LBW (<2.5 kg) | 249 (7.7) | 267 (7.5) | 364 (9.5) | 459 (10.2) | 460 (9.8) |
| | NBW (≥2.5 kg) | 2401 (73.8) | 2553 (71.9) | 3110 (80.9) | 3605 (80.1) | 3563 (75.6) |
| | HBW (≥4 kg) | 105 (3.2) | 99 (2.8) | 121 (3.1) | 156 (3.5) | 157 (3.3) |
| | Non-HBW (<4 kg) | 2545 (78.2) | 2721 (76.7) | 3353 (87.3) | 3908 (86.9) | 3866 (82.1) |
| | Missing BW | 604 (18.6) | 729 (20.5) | 368 (9.6) | 434 (9.6) | 687 (14.6) |
| Low monthly household income | <R2500 | 1737 (53.4) | 1804 (50.8) | 1660 (43.2) | 1484 (33) | 1202 (25.5) |
| | ≥R2500 | 552 (17) | 1014 (28.6) | 1686 (43.9) | 2749 (61.1) | 3109 (66) |
| Child hungry in the last year (food security)† | Never | 2148 (66) | N/A | | | |
| | Seldom | 333 (10.2) | | | | |
| | Sometimes | 583 (17.9) | | | | |
| | Often | 149 (4.6) | | | | |
| | Always | 35 (1.1) | | | | |
| Province | Eastern Cape | 437 (13.4) | 442 (12.5) | 437 (11.4) | 545 (12.1) | 545 (11.6) |
| | Free State | 163 (5) | 171 (4.8) | 200 (5.2) | 244 (5.4) | 242 (5.1) |
| | Gauteng | 274 (8.4) | 346 (9.7) | 381 (9.9) | 455 (10.1) | 538 (11.4) |
| | KwaZulu-Natal | 1057 (32.5) | 1076 (30.3) | 1188 (30.9) | 1449 (32.2) | 1534 (32.6) |
| | Limpopo | 293 (9) | 348 (9.8) | 423 (11) | 497 (11) | 471 (10) |
| | Mpumalanga | 231 (7.1) | 257 (7.2) | 283 (7.4) | 307 (6.8) | 356 (7.6) |
| | North-West | 226 (6.9) | 240 (6.8) | 269 (7) | 293 (6.5) | 296 (6.3) |
| | Northern Cape | 243 (7.5) | 224 (6.3) | 258 (6.7) | 316 (7) | 322 (6.8) |
| | Western Cape | 330 (10.1) | 344 (9.7) | 367 (9.6) | 368 (8.2) | 368 (7.8) |
| Environment | Rural formal | 324 (10) | 350 (9.9) | 343 (8.9) | 389 (8.6) | 449 (9.5) |
| | Tribal authority area | 1583 (48.6) | 1526 (43) | 1801 (46.9) | 2154 (47.9) | 2135 (45.3) |
| | Urban formal | 1133 (34.8) | 1221 (34.4) | 1319 (34.3) | 1498 (33.3) | 1702 (36.1) |
| | Urban informal | 214 (6.6) | 228 (6.4) | 257 (6.7) | 303 (6.7) | 317 (6.7) |
| Mother BMI | Underweight | 85 (2.6) | 78 (2.2) | 58 (1.5) | 98 (2.2) | 135 (2.9) |
| | Normal | 1010 (31) | 1105 (31.1) | 1250 (32.5) | 1373 (30.5) | 1485 (31.5) |
| | Overweight | 734 (22.6) | 850 (24) | 962 (25) | 1054 (23.4) | 1053 (22.4) |
| | Obese | 932 (28.6) | 987 (27.8) | 1054 (27.4) | 1377 (30.6) | 1382 (29.3) |
| | Missing | 493 (15.2) | 529 (14.9) | 518 (13.5) | 596 (13.3) | 655 (13.9) |

Continued

**Table 1** Continued

| Variable | Category | Wave 1: 2008 n (%) | Wave 2: 2010/2011 n (%) | Wave 3: 2012 n (%) | Wave 4: 2014/2015 n (%) | Wave 5: 2017 n (%) |
|---|---|---|---|---|---|---|
| Mother age | <20 | 234 (7.2) | 238 (6.7) | 259 (6.7) | 316 (7) | 322 (6.8) |
| | 20–24 | 807 (24.8) | 872 (24.6) | 971 (25.3) | 1100 (24.5) | 1062 (22.5) |
| | 25–34 | 1213 (37.3) | 1413 (39.8) | 1566 (40.8) | 1853 (41.2) | 2004 (42.5) |
| | 35–44 | 583 (17.9) | 581 (16.4) | 633 (16.5) | 682 (15.2) | 772 (16.4) |
| | 45+ | 81 (2.5) | 92 (2.6) | 82 (2.1) | 86 (1.9) | 98 (2.1) |
| | Missing | 336 (10.3) | 353 (9.9) | 331 (8.6) | 461 (10.2) | 452 (9.6) |
| Mother education | None | 131 (4) | 115 (3.2) | 76 (2) | 48 (1.1) | 81 (1.7) |
| | Primary | 505 (15.5) | 419 (11.8) | 405 (10.5) | 387 (8.6) | 97 (2.1) |
| | Secondary | 1871 (57.5) | 2265 (63.8) | 2654 (69.1) | 3176 (70.6) | 3130 (66.5) |
| | Tertiary | 132 (4.1) | 141 (4) | 172 (4.5) | 240 (5.3) | 707 (15) |
| | Missing | 615 (18.9) | 609 (17.2) | 535 (13.9) | 647 (14.4) | 695 (14.8) |

*139 misclassified or missing in 2017.
†Only included in wave 1 questionnaire.
BMI, body mass index; BW, birth weight; HBW, high birth weight; LBW, low birth weight; N/A, not applicable; NBW, normal birth weight.

## Factors associated with child nutritional status

A post-hoc sample size (power) analysis is presented in online supplementary material 10. A bivariate analysis of demographic, maternal, socioeconomic and household factors at individual nutritional status level suggests that African ethnicity (p<0.001), male gender (p=0.002), low birth weight (LBW) (p<0.001), residing in lower SES household (p<0.001), province of residence (p=0.012), lower maternal/paternal education status (p<0.001 and p=0.020, respectively) and residence in a rural/tribal authority area (p<0.001) were significantly associated with stunting (table 3). Children living in lower income households (p=0.053), lower food security (as measured through child hunger in the last year) (p<0.001), province of residence (p=0.002), having a younger mother (<20) (p=0.012) and mother having a lower BMI classification (p=0.005) were significantly associated with thinness/wasting status. Children of African ethnicity (p<0.001), higher birth weight (p=0.006), living in lower income households (p=0.001) in KwaZulu-Natal and Eastern Cape (p<0.001), as well as paternal educational attainment (p=0.033) were significantly associated with obesity status (table 3).

## DISCUSSION
### Main findings

The present study illustrates that while stunting has declined among South African children over the last 10 years, wasting and obesity appear largely unchanged, suggesting that development and public health interventions have had a variable impact. Stunting prevalence appears relatively evenly spread across South Africa, but obesity burden is more pronounced in the east of the country, whereas thinness/wasting is more pronounced in the west. In terms of progress towards WHO 2025 nutritional targets, 14 of 52 (27%) districts had an estimated wasting prevalence still exceeding 5% prevalence in 2017 as well as 17% (9/52) and 12% (6/52) districts not attaining the relative reduction in stunting prevalence required or with an increase in obesity prevalence respectively from 2012 to 2017. A further concerning pattern observed was the increasing prevalence of obesity in children under the age of 2 years. Key sociodemographic factors associated with malnutrition status were identified which likely underpins the spatial patterns (and heterogeneity) observed across the country. African children with lower birth weights residing in lower income households in rural areas with less educated mothers and fathers were particularly more likely to be stunted. Children in lower income, food-insecure households with malnourished young mothers appeared particularly more likely to be thin/wasted while African children, with higher birth weights, living in lower income households in KwaZulu-Natal and Eastern Cape were also more likely to be obese. Furthermore, low household income appeared to be positively associated with all three nutritional types. Declining childhood stunting rates from 2008 to 2017 may well have resulted from government initiatives to support food security and child health (among other things), but our findings of distinct geographic and sociodemographic variability in undernutrition and obesity rates suggest that tackling malnutrition in South Africa is complex. Models and targets for nationally driven intervention need to be carefully specified according to local environments and socioeconomic profiles.

### Contribution to existing literature

Two previous studies in South Africa among primary school-aged children dating back 25+ years (1993 and

**Table 2** Burden of stunting, thinness/wasting and obesity among children by age and survey round

| Survey wave | Age (years) | n (valid HAZ) | n (stunted) | Prop: stunted* | Estimated population stunted | n (valid BMIZ) | n (thin/ wasted) | Prop: thinness† | Estimated population thinness | n (obese) | Prop: obese‡ | Estimated population obese |
|---|---|---|---|---|---|---|---|---|---|---|---|---|
| 2008 | 0 | 220 | 31 | 0.14 (0.09, 0.22) | 153 648 (81 545, 273 371) | 180 | 21 | 0.12 (0.07, 0.2) | 133 882 (66 374, 251 867) | 32 | 0.1 (0.06, 0.15) | 107 783 (59 737, 185 749) |
| | 1 | 419 | 29 | 0.08 (0.05, 0.13) | 91 903 (48 436, 164 369) | 386 | 24 | 0.06 (0.03, 0.11) | 66 566 (29 263, 143 661) | 76 | 0.22 (0.16, 0.3) | 253 021 (159 436, 383 096) |
| | 2 | 453 | 62 | 0.15 (0.1, 0.21) | 159 241 (96 989, 250 626) | 419 | 10 | 0.03 (0.01, 0.07) | 34 613 (12 484, 87 598) | 70 | 0.14 (0.1, 0.19) | 148 357 (93 148, 227 510) |
| | 3 | 489 | 55 | 0.11 (0.08, 0.15) | 111 595 (69 906, 172 639) | 470 | 19 | 0.04 (0.02, 0.07) | 39 715 (20 205, 75 821) | 67 | 0.17 (0.12, 0.24) | 176 235 (104 092, 284 620) |
| | 4 | 498 | 48 | 0.09 (0.06, 0.13) | 93 391 (54 519, 154 136) | 461 | 25 | 0.05 (0.03, 0.08) | 52 031 (27 083, 96 623) | 34 | 0.08 (0.05, 0.12) | 80 282 (45 874, 135 732) |
| | 0–5 | 2079 | 225 | 0.11 (0.09, 0.13)§ | 591 550 (451 494, 766 049) | 1916 | 99 | 0.05 (0.04, 0.07)§ | 277 743 (196 715, 385 904) | 279 | 0.14 (0.12, 0.17)§ | 778 865 (599 156, 996 439) |
| 2010/2011 | 0 | 75 | 24 | 0.33 (0.16, 0.57) | 289 420 (114 550, 577 181) | 69 | 7 | 0.1 (0.04, 0.23) | 88 499 (30 258, 228 461) | 22 | 0.39 (0.21, 0.61) | 340 820 (153 454, 615 984) |
| | 1 | 236 | 20 | 0.06 (0.03, 0.11) | 63 995 (25 204, 132 218) | 215 | 11 | 0.07 (0.03, 0.14) | 69 776 (25 204, 173 842) | 52 | 0.29 (0.19, 0.41) | 299 127 (159 624, 499 489) |
| | 2 | 340 | 61 | 0.22 (0.16, 0.29) | 267 019 (166 414, 407 708) | 314 | 17 | 0.06 (0.03, 0.11) | 76 344 (35 363, 155 183) | 72 | 0.22 (0.16, 0.29) | 270 818 (167 454, 414 761) |
| | 3 | 427 | 52 | 0.11 (0.07, 0.16) | 130 531 (73 921, 220 389) | 402 | 20 | 0.03 (0.02, 0.06) | 39 208 (16 427, 85 938) | 78 | 0.16 (0.11, 0.23) | 195 314 (114 988, 313 258) |
| | 4 | 422 | 62 | 0.17 (0.12, 0.24) | 205 730 (122 130, 329 629) | 394 | 19 | 0.03 (0.02, 0.06) | 39 494 (17 639, 84 450) | 65 | 0.17 (0.12, 0.24) | 208 842 (126 152, 329 629) |
| | 0–5 | 1500 | 219 | 0.16 (0.13, 0.19) | 862 302 (633 920, 1 148 376) | 1394 | 74 | 0.05 (0.03, 0.07) | 265 877 (167 080, 405 309) | 289 | 0.21 (0.17, 0.26) | 1 159 133 (835 398, 1 565 968) |
| 2012 | 0 | 271 | 59 | 0.2 (0.14, 0.28) | 181 464 (108 101, 288 795) | 250 | 38 | 0.2 (0.12, 0.3) | 179 118 (95 658, 311 389) | 55 | 0.19 (0.12, 0.28) | 169 192 (94 880, 284 482) |
| | 1 | 544 | 78 | 0.13 (0.09, 0.17) | 132 310 (80 796, 207 206) | 538 | 27 | 0.08 (0.05, 0.13) | 80 862 (40 842, 150 046) | 138 | 0.23 (0.18, 0.28) | 234 062 (157 153, 334 626) |
| | 2 | 629 | 72 | 0.1 (0.07, 0.14) | 116 230 (68 690, 187 924) | 629 | 49 | 0.05 (0.03, 0.07) | 55 866 (30 861, 97 391) | 147 | 0.23 (0.18, 0.29) | 269 508 (176 205, 392 309) |
| | 3 | 710 | 82 | 0.11 (0.08, 0.16) | 142 259 (82 987, 232 297) | 692 | 29 | 0.03 (0.02, 0.06) | 43 898 (20 928, 87 296) | 102 | 0.15 (0.11, 0.2) | 191 943 (117 798, 297 399) |
| | 4 | 771 | 112 | 0.16 (0.12, 0.2) | 221 293 (142 258, 330 201) | 762 | 30 | 0.03 (0.0, 0.05) | 43 556 (20 731, 87 406) | 118 | 0.18 (0.14, 0.22) | 250 658 (167 278, 362 573) |
| | 0–5 | 2925 | 403 | 0.13 (0.11, 0.16) | 762 303 (567 517, 1 001 855) | 2871 | 173 | 0.06 (0.05, 0.07) | 328 768 (230 074, 458 914) | 560 | 0.19 (0.17, 0.22) | 1 112 487 (853 832, 1 415 525) |
| 2014/2015 | 0 | 434 | 74 | 0.12 (0.08, 0.18) | 144 201 (81 319, 240 730) | 421 | 37 | 0.1 (0.06, 0.18) | 123 211 (59 233, 240 730) | 78 | 0.17 (0.12, 0.23) | 197 209 (117 461, 313 223) |
| | 1 | 801 | 53 | 0.06 (0.04, 0.08) | 67 916 (39 433, 112 566) | 801 | 24 | 0.03 (0.01, 0.08) | 39 657 (9858, 101 845) | 169 | 0.23 (0.18, 0.28) | 266 780 (179 421, 379 240) |
| | 2 | 785 | 65 | 0.08 (0.05, 0.12) | 85 985 (48 668, 146 305) | 781 | 16 | 0.02 (0.01, 0.03) | 16 222 (6309, 39 015) | 128 | 0.16 (0.12, 0.22) | 170 803 (106 348, 263 349) |

Continued

**Table 2** Continued

| Survey wave | Age (years) | n (valid HAZ) | n (stunted) | Prop: stunted* | Estimated population stunted | n (valid BMIZ) | n (thin/ wasted) | Prop: thinness† | Estimated population thinness | n (obese) | Prop: obese‡ | Estimated population obese |
|---|---|---|---|---|---|---|---|---|---|---|---|---|
| | 3 | 853 | 82 | 0.08 (0.06, 0.11) | 89857 (54478, 143034) | 845 | 24 | 0.04 (0.02, 0.07) | 40865 (18323, 86890) | 79 | 0.12 (0.08, 0.15) | 133857 (83637, 205862) |
| | 4 | 899 | 67 | 0.06 (0.04, 0.09) | 77887 (45801, 127320) | 897 | 19 | 0.02 (0.01, 0.05) | 30376 (12301, 71898) | 56 | 0.06 (0.04, 0.11) | 82300 (38662, 166265) |
| | 0–5 | 3772 | 341 | 0.08 (0.06, 0.09) | 441281 (327611, 581707) | 3745 | 120 | 0.04 (0.03, 0.05) | 213012 (130004, 333338) | 510 | 0.14 (0.12, 0.17) | 834444 (618820, 1098053) |
| 2017 | 0 | 372 | 50 | 0.13 (0.08, 0.19) | 125347 (68160, 218303) | 357 | 32 | 0.12 (0.07, 0.2) | 121396 (62270, 221478) | 70 | 0.18 (0.12, 0.25) | 174538 (104344, 278066) |
| | 1 | 760 | 55 | 0.08 (0.05, 0.11) | 95527 (56435, 153804) | 742 | 23 | 0.03 (0.02, 0.07) | 42416 (17767, 94222) | 146 | 0.23 (0.19, 0.29) | 285123 (194388, 403216) |
| | 2 | 833 | 63 | 0.07 (0.05, 0.11) | 94807 (54147, 158550) | 830 | 20 | 0.03 (0.02, 0.07) | 43976 (18786, 99279) | 130 | 0.15 (0.12, 0.19) | 191812 (127079, 280056) |
| | 3 | 875 | 77 | 0.08 (0.05, 0.12) | 99890 (54439, 175689) | 872 | 14 | 0.02 (0.01, 0.06) | 30726 (10888, 79204) | 77 | 0.07 (0.05, 0.1) | 88889 (54439, 138247) |
| | 4 | 900 | 59 | 0.05 (0.04, 0.07) | 57363 (34849, 91231) | 899 | 23 | 0.03 (0.01, 0.05) | 29923 (13628, 62962) | 47 | 0.06 (0.04, 0.08) | 63912 (36990, 105365) |
| | 0–5 | 3740 | 304 | 0.08 (0.06, 0.09)§ | 445295 (326192, 593240) | 3700 | 112 | 0.04 (0.03, 0.05)§ | 223236 (136790, 345514) | 470 | 0.13 (0.11, 0.15)§ | 758650 (583989, 964831) |
| At last observation | 0–5 | 10711 | 1049 | 0.09 (0.08, 0.10) | 1397020 (1177247, 1616793) | 10467 | 391 | 0.04 (0.03, 0.05) | 560806 (448656, 672957) | 1438 | 0.14 (0.13, 0.16) | 2048650 (1722242, 2375058) |

*HAZ ≤−2 SD.
†BMI-for-age z-score ≤−2 SD.
‡BMI-for-age z-score ≥+2 SD.
§Significance tests (survey weighted logistic regression) among children 0–5: stunting (2017 vs 2008) p=0.007; thinness/wasting (2017 vs 2008) p=0.131; obesity (2017 vs 2008) p=0.312.
BMI, body mass index; BMIZ, BMI-for-age z-score; HAZ, height-for-age z-score.

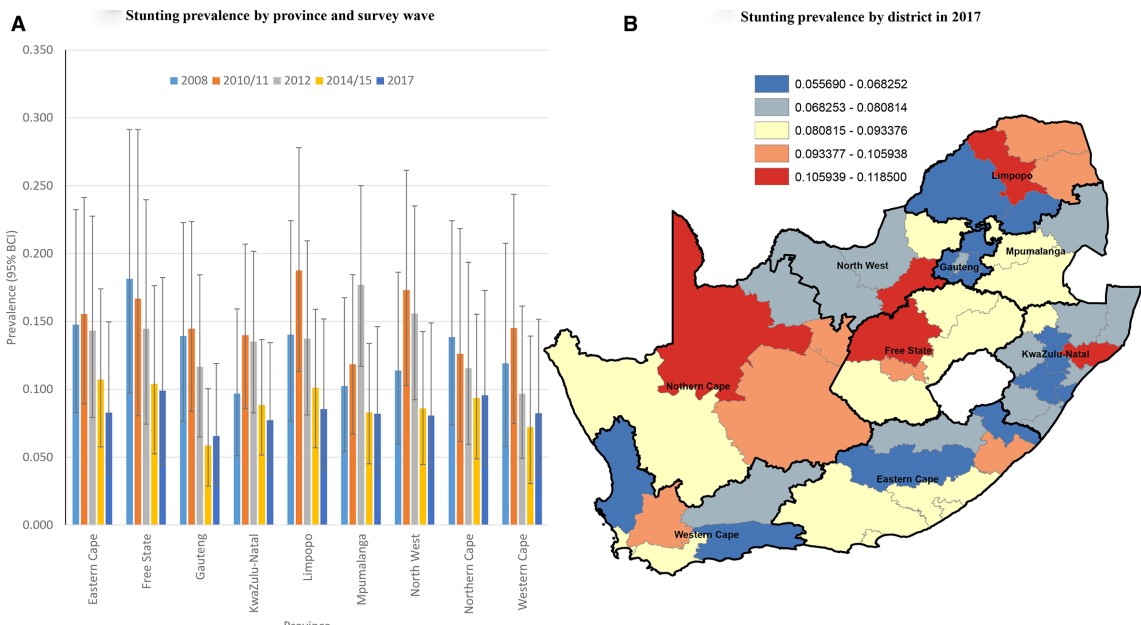

**Figure 1** Bayesian posterior median smoothed prevalence of stunting by province (and wave, A) and district-level prevalence (equal intervals, 2017, B) among children <5 years. BCI, Bayesian credibility interval.

1994, respectively) used cross-sectional data,[26 27] thus limiting insight into temporal trends. Furthermore, the study by Jinabhai *et al*[27] was restricted to KwaZulu-Natal limiting national representativeness. Another cross-sectional study in South Africa in 2001–2003 among primary school children in five South African provinces suggested that relative to 1993 prevalence of undernutrition had decreased while obesity had increased.[27 28] Thus, these previous data are now outdated, were largely focused on primary school-aged children, as well as cross sectional in nature and geographically restricted.

This is also the first spatial-temporal Bayesian-shared component analysis of malnutrition trends among children in South Africa using geographically representative repeated panel data over a 10-year period. The current study focusing on children under 5 years of age suggests that there is prominent geographic heterogeneity in malnutrition burden in South Africa in this youngest age group. This is in line with findings from other settings in Africa that have documented similar spatial heterogeneity[29] and persistence of these malnutrition inequalities has been demonstrated in

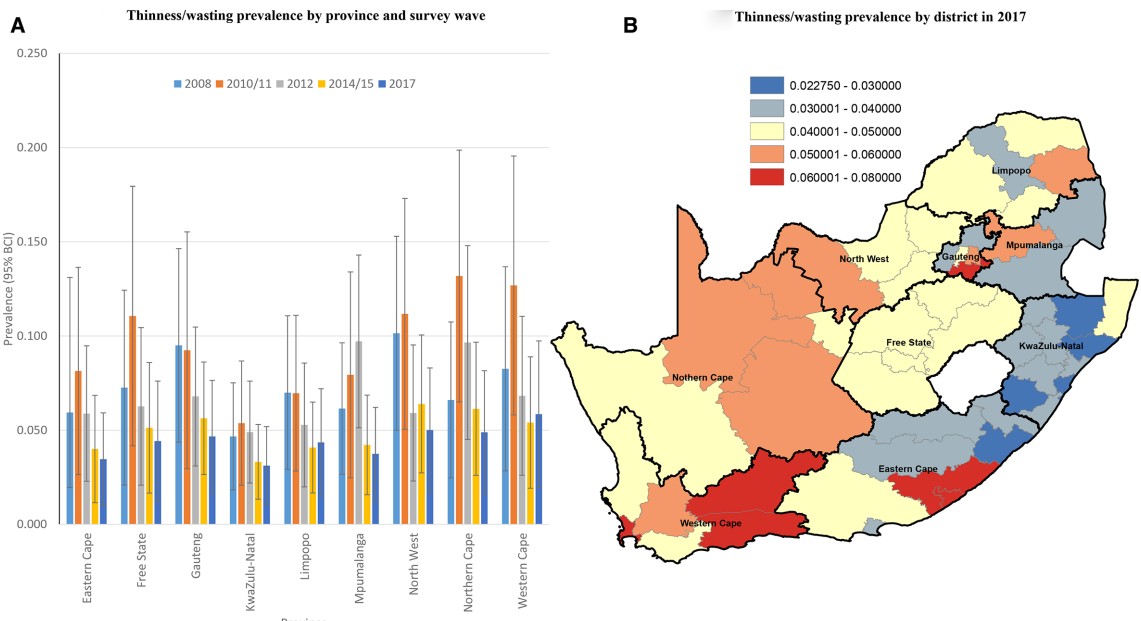

**Figure 2** Bayesian posterior median smoothed prevalence of thinness/wasting by province (and wave, A) and district-level prevalence (equal intervals, 2017, B) among children <5 years. BCI, Bayesian credibility interval.

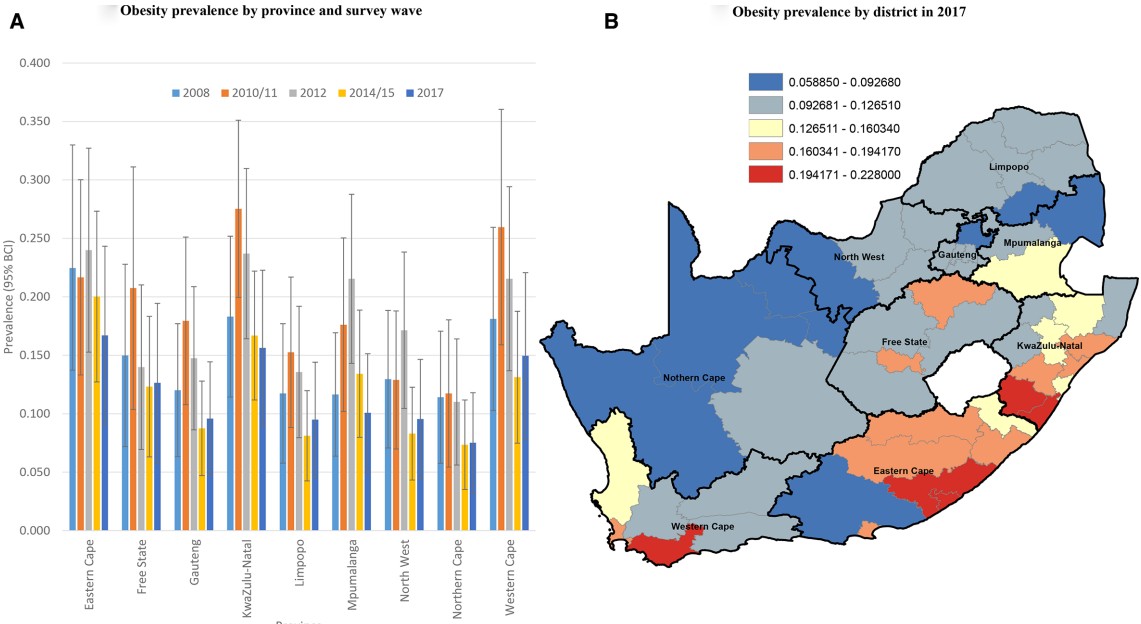

**Figure 3** Bayesian posterior median smoothed prevalence of obesity by province (and wave, A) and district-level prevalence (equal intervals, 2017, B) among children <5 years. BCI, Bayesian credibility interval.

an 80-country study further highlighting this ongoing public health conundrum.[30 31] Our results demonstrate a strong west to east gradient of higher underweight burden on the western side of South Africa and greater obesity on the eastern seaboard (Eastern Cape and KwaZulu-Natal). A map of poverty and inequality in South Africa (https://southafrica-info.com/people/mapping-poverty-in-south-africa/) illustrates the coexistence of high levels of poverty and inequality in many parts of KwaZulu-Natal and the Eastern Cape with high levels of overweight/obesity. This is further confirmed by our individual child-level analysis which suggested a significantly higher obesity prevalence in lower income households. Metropolitan areas displayed high levels of nutritional inequality that complement national studies of poverty and inequality.[32]

Undernutrition and overnutrition status appeared positively associated with lower household income classification. This finding of stunting and wasting disproportionately affecting the poor has been often demonstrated.[33] Other studies in Africa in particular have documented similar patterns, that is, children living in low SES households, children who live in peripheral areas and whose mothers had little or no schooling were at significantly higher risk of malnutrition.[34] The inconsistent challenges facing health authorities are occurring in the face of rapid urbanisation and industrialisation that simultaneously attract both the rich and the poor to live in the same geographic districts.[35] The heterogeneous geographic relationship between household income and undernutrition is also affected by the allocation of household income that is a function of maternal education, access to markets, infrastructure and sanitation.[36] Additionally, these data suggest that there is a strong and highly significant

association between higher food insecurity (child hunger frequency in the preceding year) and increased thinness/wasting. Community and government-based packages of support need to be highly targeted to the poorest and most food-insecure households to further reduce inequality in this regard and maximise reductions in malnutrition.

Our findings suggest that children with LBW (due to preterm delivery, fetal/intrauterine growth restriction or a combination of the two) were significantly more likely to be stunted than normal weight babies and this has been demonstrated in many other low and middle-income settings[37]. SES/factors are known risk factors for LBW[38] and may in part explain the significant association found between stunting and lower household income. South Africa has the higher number of incident and prevalent HIV infections globally.[39] A further important contextual risk factor for LBW is maternal HIV status. A systematic review and large observational studies focusing on low and middle-income countries (LMIC) suggest a strong and significant association between maternal HIV infection and LBW.[40 41] Evidence from South Africa also suggests the anthropometric z-score of HIV-infected children appears to be consistently lower when compared with HIV-exposed but uninfected children.[42] We also observed a significantly higher prevalence of stunting among male children which has been demonstrated previously in a meta-analysis for SSA,[43] the suggested cause of which might be that male children are more vulnerable to health inequalities relative to female children of the same age. Strengthening community-based packages of care and community health worker (CHW) performance/skills in rural and high-burden geographies are key strategies to improve primary

**Table 3** Demographic, socioeconomic and maternal factors associated with nutritional status among children under 5 years, 2008–2017

| Variable | Category | Stunted Yes (% col) | No (% col) | P value | Thin/wasted Yes (% col) | No (% col) | P value | Obese Yes (% col) | No (% col) | P value |
|---|---|---|---|---|---|---|---|---|---|---|
| Ethnicity | African | 0.939 (0.9027, 0.9619) | 0.871 (0.8284, 0.9039) | <0.001 | 0.885 (0.8155, 0.9306) | 0.879 (0.8383, 0.9108) | 0.823 | 0.931 (0.9017, 0.9522) | 0.870 (0.8255, 0.9044) | <0.001 |
| | Coloured | 0.053 (0.0311, 0.0879) | 0.074 (0.0486, 0.1116) | | 0.076 (0.0415, 0.1344) | 0.072 (0.0474, 0.1081) | | 0.052 (0.0333, 0.0789) | 0.076 (0.0495, 0.1152) | |
| | Asian/Indian | 0.003 (4.0e-04, 0.0202) | 0.012 (0.0049, 0.0294) | | 0.015 (0.0026, 0.077) | 0.011 (0.0046, 0.0278) | | 0.004 (8.4e-04, 0.0141) | 0.013 (0.0053, 0.0317) | |
| | White | 0.006 (0.0017, 0.0179) | 0.039 (0.0238, 0.0627) | | 0.025 (0.0083, 0.0711) | 0.037 (0.0229, 0.0605) | | 0.014 (0.0066, 0.0287) | 0.041 (0.0248, 0.067) | |
| Gender | Male | 0.562 (0.5204, 0.603) | 0.496 (0.4797, 0.5121) | 0.002 | 0.514 (0.4543, 0.5742) | 0.501 (0.4846, 0.5182) | 0.686 | 0.523 (0.488, 0.5575) | 0.498 (0.481, 0.5151) | 0.178 |
| | Female | 0.438 (0.397, 0.4796) | 0.504 (0.4879, 0.5203) | | 0.486 (0.4258, 0.5457) | 0.499 (0.4818, 0.5154) | | 0.477 (0.4425, 0.512) | 0.502 (0.4849, 0.519) | |
| Birth weight | LBW (<2.5 kg) | 0.148 (0.1143, 0.1891) | 0.098 (0.0849, 0.1117) | <0.001 | 0.13 (0.0891, 0.1867) | 0.098 (0.0858, 0.1111) | 0.163 | 0.072 (0.0554, 0.0938) | 0.104 (0.0919, 0.118) | 0.006 |
| | NBW (≥2.5 kg) | 0.852 (0.8109, 0.8857) | 0.903 (0.8883, 0.9151) | | 0.87 (0.8133, 0.9109) | 0.902 (0.8889, 0.9142) | | 0.928 (0.9062, 0.9446) | 0.896 (0.882, 0.9081) | |
| | HBW (≥4 kg) | Not applicable | | | Not applicable | | | 0.056 (0.0419, 0.0751) | 0.04 (0.0323, 0.0496) | 0.037 |
| | Non-HBW (<4 kg) | | | | | | | 0.944 (0.9249, 0.9581) | 0.96 (0.9504, 0.9677) | |
| Income quantile | Lowest | 0.294 (0.2567, 0.3334) | 0.199 (0.1824, 0.2156) | <0.001 | 0.234 (0.1805, 0.2973) | 0.203 (0.1872, 0.2195) | 0.481 | 0.226 (0.1936, 0.2617) | 0.2 (0.1834, 0.2181) | 0.422 |
| | Low | 0.205 (0.1714, 0.2423) | 0.187 (0.1714, 0.2028) | | 0.214 (0.1698, 0.2656) | 0.188 (0.173, 0.2029) | | 0.203 (0.1725, 0.2377) | 0.186 (0.1723, 0.2005) | |
| | Middle | 0.183 (0.1555, 0.2148) | 0.200 (0.1853, 0.2154) | | 0.169 (0.1305, 0.2167) | 0.201 (0.1871, 0.2162) | | 0.18 (0.1501, 0.2135) | 0.204 (0.1891, 0.2189) | |
| | High | 0.197 (0.1579, 0.243) | 0.186 (0.1714, 0.2021) | | 0.184 (0.1394, 0.2377) | 0.191 (0.1751, 0.2074) | | 0.182 (0.1445, 0.2269) | 0.192 (0.1769, 0.2079) | |
| | Highest | 0.122 (0.0924, 0.1583) | 0.229 (0.2015, 0.2585) | | 0.2 (0.1494, 0.2612) | 0.218 (0.1906, 0.2476) | | 0.209 (0.1673, 0.2586) | 0.218 (0.1915, 0.2478) | |
| Low monthly household income | <R2500 | 0.566 (0.5213, 0.6101) | 0.417 (0.3929, 0.4409) | <0.001 | 0.488 (0.4228, 0.5544) | 0.423 (0.3994, 0.4469) | 0.053 | 0.481 (0.4406, 0.5214) | 0.416 (0.392, 0.4396) | 0.001 |
| | ≥R2500 | 0.434 (0.3899, 0.4787) | 0.583 (0.5591, 0.6071) | | 0.512 (0.4456, 0.5772) | 0.577 (0.5531, 0.6006) | | 0.519 (0.4786, 0.5594) | 0.584 (0.5604, 0.608) | |

**Table 3** Continued

| Variable | Category | Stunted | | | Thin/wasted | | | Obese | | |
|---|---|---|---|---|---|---|---|---|---|---|
| | | Yes (% col) | No (% col) | P value | Yes (% col) | No (% col) | P value | Yes (% col) | No (% col) | P value |
| Child hungry in the last year (food security)* | Never | 0.689 (0.595, 0.7701) | 0.697 (0.6568, 0.7346) | 0.505 | 0.512 (0.3895, 0.6337) | 0.704 (0.6643, 0.7401) | <0.001 | 0.707 (0.6302, 0.773) | 0.693 (0.6522, 0.7318) | 0.645 |
| | Seldom | 0.127 (0.0669, 0.2286) | 0.096 (0.0766, 0.1193) | | 0.111 (0.056, 0.2074) | 0.097 (0.0765, 0.1219) | | 0.076 (0.0499, 0.1138) | 0.102 (0.0787, 0.13) | |
| | Sometimes | 0.126 (0.0807, 0.1919) | 0.155 (0.1303, 0.184) | | 0.317 (0.219, 0.4354) | 0.148 (0.1243, 0.1752) | | 0.154 (0.0994, 0.231) | 0.155 (0.1316, 0.1822) | |
| | Often | 0.054 (0.0265, 0.1049) | 0.043 (0.0276, 0.0653) | | 0.052 (0.0222, 0.1181) | 0.042 (0.0272, 0.0655) | | 0.052 (0.0272, 0.0981) | 0.041 (0.0269, 0.0621) | |
| | Always | 0.004 (0.0011, 0.0144) | 0.009 (0.0048, 0.0173) | | 0.007 (0.001, 0.0504) | 0.009 (0.0049, 0.0171) | | 0.011 (0.0039, 0.0313) | 0.009 (0.0048, 0.016) | |
| Province | Eastern Cape | 0.165 (0.1137, 0.2336) | 0.132 (0.0978, 0.1765) | 0.012 | 0.075 (0.0492, 0.1137) | 0.137 (0.1007, 0.1838) | 0.002 | 0.19 (0.1321, 0.2643) | 0.124 (0.0916, 0.1652) | <0.001 |
| | Free State | 0.066 (0.0441, 0.0961) | 0.050 (0.036, 0.0678) | | 0.032 (0.0169, 0.0611) | 0.052 (0.0376, 0.0709) | | 0.045 (0.0298, 0.068) | 0.052 (0.0379, 0.071) | |
| | Gauteng | 0.188 (0.132, 0.2606) | 0.236 (0.1819, 0.2996) | | 0.298 (0.1952, 0.4272) | 0.231 (0.1784, 0.2937) | | 0.173 (0.1234, 0.2365) | 0.246 (0.1891, 0.3128) | |
| | KwaZulu-Natal | 0.218 (0.1619, 0.2857) | 0.227 (0.1801, 0.2819) | | 0.161 (0.1151, 0.2195) | 0.228 (0.1804, 0.2835) | | 0.293 (0.217, 0.3834) | 0.212 (0.1691, 0.262) | |
| | Limpopo | 0.143 (0.0947, 0.2088) | 0.109 (0.0816, 0.1444) | | 0.129 (0.0823, 0.195) | 0.113 (0.0842, 0.1491) | | 0.074 (0.0514, 0.105) | 0.121 (0.0902, 0.1599) | |
| | Mpumalanga | 0.085 (0.0541, 0.1318) | 0.083 (0.0621, 0.1102) | | 0.096 (0.0611, 0.1487) | 0.082 (0.0611, 0.1098) | | 0.074 (0.0506, 0.1079) | 0.085 (0.0626, 0.1131) | |
| | North-West | 0.055 (0.0355, 0.0833) | 0.05 (0.035, 0.0709) | | 0.06 (0.0376, 0.0943) | 0.05 (0.0346, 0.0712) | | 0.038 (0.0252, 0.056) | 0.053 (0.0362, 0.076) | |
| | Northern Cape | 0.022 (0.0141, 0.0333) | 0.023 (0.0163, 0.031) | | 0.033 (0.0217, 0.0489) | 0.022 (0.0159, 0.0303) | | 0.011 (0.0072, 0.0156) | 0.025 (0.0178, 0.0341) | |
| | Western Cape | 0.06 (0.0321, 0.1089) | 0.091 (0.0606, 0.134) | | 0.116 (0.0638, 0.2016) | 0.086 (0.0572, 0.1262) | | 0.103 (0.0626, 0.1641) | 0.084 (0.0554, 0.1254) | |
| Environment | Rural/tribal authority | 0.519 (0.4417, 0.5963) | 0.451 (0.3933, 0.5091) | <0.001 | 0.429 (0.3428, 0.5201) | 0.46 (0.4021, 0.5193) | 0.647 | 0.466 (0.3857, 0.5479) | 0.457 (0.4002, 0.5158) | 0.111 |
| | Urban informal | 0.122 (0.0737, 0.1943) | 0.101 (0.0628, 0.1592) | | 0.1 (0.0557, 0.1743) | 0.102 (0.0636, 0.161) | | 0.133 (0.0691, 0.239) | 0.097 (0.0618, 0.148) | |
| | Urban formal | 0.359 (0.292, 0.4319) | 0.448 (0.389, 0.509) | | 0.47 (0.3734, 0.5696) | 0.437 (0.3787, 0.4979) | | 0.402 (0.3261, 0.4821) | 0.446 (0.3868, 0.5066) | |

Continued

**Table 3** Continued

| Variable | Category | Stunted Yes (% col) | No (% col) | P value | Thin/wasted Yes (% col) | No (% col) | P value | Obese Yes (% col) | No (% col) | P value |
|---|---|---|---|---|---|---|---|---|---|---|
| Mother BMI | Underweight | 0.041 (0.0271, 0.0604) | 0.022 (0.0178, 0.0282) | **0.003** | 0.051 (0.0298, 0.0867) | 0.023 (0.018, 0.0281) | **0.005** | 0.019 (0.01, 0.0351) | 0.025 (0.0198, 0.0311) | 0.135 |
| | Normal | 0.397 (0.3521, 0.444) | 0.344 (0.3213, 0.3683) | | 0.418 (0.3455, 0.4946) | 0.348 (0.3251, 0.3724) | | 0.327 (0.2853, 0.3708) | 0.356 (0.332, 0.3815) | |
| | Overweight | 0.268 (0.2311, 0.3092) | 0.273 (0.2565, 0.289) | | 0.249 (0.199, 0.3064) | 0.272 (0.2565, 0.2881) | | 0.26 (0.23, 0.2922) | 0.273 (0.2567, 0.2899) | |
| | Obese | 0.294 (0.2452, 0.3479) | 0.361 (0.3342, 0.3882) | | 0.282 (0.2137, 0.3615) | 0.357 (0.3298, 0.3853) | | 0.395 (0.3514, 0.4396) | 0.346 (0.3175, 0.3753) | |
| Mother age | <20 | 0.073 (0.0562, 0.0947) | 0.048 (0.0419, 0.0555) | 0.156 | 0.112 (0.0574, 0.206) | 0.047 (0.041, 0.0532) | **0.007** | 0.057 (0.0456, 0.0701) | 0.049 (0.0418, 0.0562) | 0.121 |
| | 20–24 | 0.219 (0.1852, 0.2571) | 0.230 (0.2152, 0.2459) | | 0.258 (0.201, 0.3252) | 0.23 (0.2138, 0.2461) | | 0.265 (0.2272, 0.3069) | 0.224 (0.2091, 0.2405) | |
| | 25–34 | 0.468 (0.4191, 0.5175) | 0.491 (0.4705, 0.5107) | | 0.398 (0.3311, 0.4691) | 0.492 (0.4713, 0.5118) | | 0.472 (0.425, 0.5189) | 0.49 (0.4691, 0.511) | |
| | 35–44 | 0.215 (0.1731, 0.2638) | 0.210 (0.191, 0.2297) | | 0.213 (0.1682, 0.2667) | 0.211 (0.1923, 0.2301) | | 0.191 (0.1536, 0.236) | 0.214 (0.1952, 0.2348) | |
| | 45+ | 0.025 (0.0161, 0.0381) | 0.021 (0.0177, 0.0256) | | 0.019 (0.0089, 0.0388) | 0.022 (0.018, 0.0261) | | 0.015 (0.0095, 0.024) | 0.023 (0.0186, 0.0278) | |
| Mother education | None | 0.023 (0.0136, 0.0397) | 0.018 (0.0144, 0.0226) | **<0.001** | 0.025 (0.0127, 0.0479) | 0.019 (0.0148, 0.0239) | **0.001** | 0.025 (0.0157, 0.0406) | 0.018 (0.014, 0.023) | 0.568 |
| | Primary | 0.121 (0.0921, 0.1576) | 0.072 (0.0625, 0.0835) | | 0.132 (0.095, 0.1804) | 0.071 (0.061, 0.0825) | | 0.067 (0.0488, 0.0925) | 0.075 (0.0647, 0.0869) | |
| | Secondary | 0.799 (0.7529, 0.8385) | 0.796 (0.7777, 0.8134) | | 0.715 (0.6506, 0.7712) | 0.802 (0.7832, 0.8203) | | 0.803 (0.7595, 0.8398) | 0.798 (0.7785, 0.8152) | |
| | Tertiary | 0.057 (0.0364, 0.0868) | 0.114 (0.0985, 0.1307) | | 0.129 (0.0862, 0.1874) | 0.108 (0.0925, 0.1251) | | 0.105 (0.077, 0.1405) | 0.11 (0.0938, 0.1275) | |
| Father education | None | 0.003 (8.0e-04, 0.0082) | 0.003 (0.0017, 0.0051) | **0.020** | 0.005 (6.7e-04, 0.0333) | 0.003 (0.0017, 0.0051) | 0.960 | 0.002 (6.8e-04, 0.0053) | 0.003 (0.0018, 0.0057) | **0.033** |
| | Primary | 0.646 (0.5533, 0.7282) | 0.56 (0.5162, 0.6028) | | 0.565 (0.4542, 0.6703) | 0.556 (0.5118, 0.5984) | | 0.584 (0.499, 0.6637) | 0.551 (0.505, 0.5971) | |
| | Secondary | 0.275 (0.2008, 0.3629) | 0.389 (0.3468, 0.4334) | | 0.382 (0.2783, 0.4965) | 0.387 (0.3448, 0.431) | | 0.318 (0.2475, 0.3976) | 0.398 (0.3529, 0.445) | |
| | Tertiary | 0.077 (0.0413, 0.1403) | 0.048 (0.035, 0.0651) | | 0.048 (0.0206, 0.1099) | 0.055 (0.0389, 0.0761) | | 0.097 (0.0502, 0.1779) | 0.047 (0.0338, 0.0658) | |

Statistically significant associations highlighted in bold.

*Only included in wave 1 questionnaire.

BMI, body mass index; HBW, high birth weight; LBW, low birth weight; NBW, normal birth weight.

healthcare delivery through better identification of women at higher risk of poor birth outcomes (eg, HIV positive, history of previous poor birth outcomes and/ or currently malnourished), higher referral rates for facility births and improved linkage to other health as well as social services.[44] Lastly, given the high adolescent fertility rates in many parts of South Africa,[45] there is also much scope to improve CHW identification of households with higher risk malnourished adolescent girls prior to pregnancy to ensure more optimal linkage to government and social support to ensure adequate nutrition as well as improved awareness regarding family planning practices, for example, ensuring adequate birth spacing.[46]

Obesity in children has a complex aetiology that includes a wide range of socioeconomic, demographic, environmental and cultural variables,[47] such as household composition, mother's education, household income, household size, environmental factors, rural versus urban location and sanitation.[9 48] The high burden of obesity is likely associated with a progressive increase in the per-capita food supply and consumption of high-calorific foods (eg, fat, sugar, fast and/ or processed foods) in South Africa.[49] This rapidly changing dietary pattern has, in part, been attributed to urbanisation, growing and expanding supermarkets/formal food retailers and the availability of fast/ processed foods.[50] An interesting finding in these data was the significant positive association between child obesity status and residing in a lower income household. This association has been demonstrated previously[51–53] and this evidence base is growing. This conforms with the idea that lower and higher income households/ families often have a higher obesity risk than middle-income households, that is, so-called U-shaped association. Lower income or economically deprived families often replace health fresh food options with cheaper and more calorific processed foods.[52] Multiple studies have demonstrated that the majority of low-income South Africans have a low dietary diversity, and, therefore, consume a limited food range consisting predominantly of a starchy staple such as bread and maize, with low intakes of vegetables and fruits.[49] Future work will characterise food purchasing patterns (and changes over time) among households in South Africa which will be compared with paired longitudinal anthropometric measurements to identify specific dietary patterns associated with child nutritional status.

Lastly and contextually, body mass is culturally influenced in South Africa, and the high level of obesity in KwaZulu-Natal and Eastern Cape may at least in part be a result of cultural beliefs that associate overweight with wealth and good health.[54] Geographic patterns of higher obesity in South Africa appeared to overlap areas of high poverty particularly on the eastern side of the country (https://southafrica-info.com/people/ mapping-poverty-in-south-africa/) and thus not solely concentrated among higher socioeconomic households.

## Strengths

To our knowledge this is the first spatial-temporal analysis of malnutrition trends among children under 5 years of age in South Africa. We used standardised anthropometric measurements of children and their mothers from nationally representative repeated panel data over a 10-year period. The panel nature of the design allows assessment of change in malnutrition burden within the same individuals/households observed at multiple time points. A further strength was the implementation of a fully Bayesian space-time shared component model to produce more stable joint estimates of malnutrition by province, district and year.

## Weaknesses

The study has several limitations. First, missing or invalid weight/height measurements (especially in wave 2, and among infants—online supplementary material 7) may have introduced selection bias (if not missing at random), and may thus have affected both the internal validity and the representativeness of the findings in the broader South African context. Second, as the primary panel study was not designed/powered for provincial[14] and lower geographic-level analysis, we cannot discount the resultant impact on precision/random variability when analysing at provincial/district level (administrative tier just below the province) and further stratification by sociodemographic correlates. Third, we cannot discount the effect of interobserver variability across different study districts, despite extensive interviewer training and standardisation of study protocols. All anthropometric measurements (eg, weight, height) were taken in duplicate in NIDS[26] which would have ensured better reliability.

## Cost of malnutrition, policy and research needs

Estimating the cost of child malnutrition in South Africa is extremely complicated and no locally determined cost data exist. Data from the USA suggest that the incremental lifetime direct medical cost for a 10-year-old obese child relative to a 10-year-old normal weight child ranges from US$12 660 to US$19 630.[55] Estimates of the cost of treating wasted children are approximately US$200 per child[56] while stunting has been consistently linked to worse economic outcomes in adulthood[57] and estimates suggest that, on average, the future per-capita income penalty for a stunted individual could be as large as 9%–10% in SSA.[58] Urgent investments are needed to accelerate the reduction of all forms of malnutrition, as well as to curb the obesity epidemic among young children in South Africa. There is also considerable evidence that indicates childhood wasting and stunting can be reduced by 60% and 20%, respectively, using 10 nutrition-specific interventions,[59] with an estimated return on investment of 18:1, that is, for US$1 spent on implementing effective programmes there would be US$18 return in future economic benefits.[60] Very few obesity prevention interventions targeting children have been effective and a comprehensive multifaceted strategy

tackling diet, physical inactivity, coupled with psycho-social support and local food environment change may prove more effective. Nutrition policies tackling child obesity must promote household nutrition security and healthy growth, decrease overconsumption of nutrient-poor foods, better shield children from increasingly pervasive marketing of energy-dense, nutrient-poor foods and sugar-sweetened beverages as well as reduction of growing physical inactivity.[61]

Our findings suggest the need to implement evidence-based child health strategies and policy (eg, further social grant support to vulnerable and impoverished households) that is tailored to specific geographies and socially disadvantaged subpopulations. A higher prevalence of child thinness/wasting among younger mothers (<25) in poorer, food-insecure household highlights the importance of policies that enable younger mothers to adequately care for their children in all settings. Integrated nutrition programmes in LMICs have had a substantial impact on child nutrition and health via a combination of multisector-targeted interventions.[62] Furthermore, implementation and/or strengthening of school-based food programme can provide a launching pad for preventive programmes, including education and awareness, provision of healthier/more nutrition food options and micronutrient supplementation, deworming, increased immunisation coverage and improved growth monitoring as well as counselling.[62] This may be especially true of obese children where high prevalence was observed in higher income households with higher food purchasing power and where local food environments are likely to be an important contextual determinant. A further contextual trend which may further compound this problem is the rapidly rising median household income observed over the period (from ZAR1400 in 2008 to ZAR3640 by 2017).

## CONCLUSIONS

The heterogeneity of malnutrition is a feature of spatial inequality and rapid urbanisation that has manifested in widening levels of inequality in South Africa's districts and a need to reassess where nutrition programmes need to be further decentralised to the highest risk municipalities and local communities to maximise effectiveness. This work provides the first district-level ranking of childhood overweight, thinness/wasting and stunting and allows a differentiated proactive tailored intervention to be developed for each municipal district. The dual epidemic of undernutrition and overweight/obesity requires differential geographical policy inputs in metropolitan areas and districts across the rural-urban divide. The current and future health cost of malnutrition among South African children is likely substantial based on previous costing estimates. There is an urgent need to address nutrition problems among preschool-aged children in South Africa and other LMICs. Effective public health planning and

geographically/contextually tailored interventions are required at subnational level to address this challenge. The analytical framework employed in this study we believe will have definite utility in other settings.

**Author affiliations**
[1]Department of Disease Control, Faculty of Infectious and Tropical Diseases, London School of Hygiene and Tropical Medicine, London, UK
[2]Department of Public Health Medicine, School of Nursing and Public Health, University of KwaZulu-Natal, Durban, South Africa
[3]Department of Health Metrics Sciences, School of Medicine, University of Washington, Seattle, Washington, USA
[4]Department of Population Health, Faculty of Epidemiology and Population Health, London School of Hygiene and Tropical Medicine, London, UK
[5]KwaZulu-Natal Department of Health, South African Government Department, Durban, South Africa
[6]College of Social Science, University of Lincoln, Lincoln, UK
[7]Africa Health Research Institute, Durban, KwaZulu-Natal, South Africa
[8]School of Life Sciences, University of KwaZulu-Natal, Durban, KwaZulu-Natal, South Africa

**Contributors** BS contributed to the conceptualisation/design, methodology and data analysis, drafted the initial manuscript and approved the final version of the manuscript. ADD and RS participated in funding acquisition, conceptualisation/design and supervision. KS, RG, EL, PS, FT, ADD and RS reviewed/edited the manuscript for critically important intellectual content and approved the final version of the manuscript.

**Funding** This study forms part of the Sustainable and Healthy Food Systems (SHEFS) project supported by the Wellcome Trust's Our Planet, Our Health programme (grant number 205200/Z/16/Z).

**Disclaimer** The funders of the study had no role in study design, data collection, data analysis, data interpretation or writing of the report.

**Map disclaimer** The depiction of boundaries on the map(s) in this article do not imply the expression of any opinion whatsoever on the part of BMJ (or any member of its group) concerning the legal status of any country, territory, jurisdiction or area or of its authorities. The map(s) are provided without any warranty of any kind, either express or implied.

**Competing interests** None declared.

**Patient consent for publication** Not required.

**Ethics approval** Approval for the primary study was granted by the Ethics Committee of the University of Cape Town. The current study is a secondary data analysis of an open access data set and does not require further ethical approval

**Provenance and peer review** Not commissioned; externally peer reviewed.

**Data availability statement** Data are available in a public, open access repository. Data from the South African National Income Dynamics Study (SA-NIDS) are available on the following open access repository: https://www.datafirst.uct.ac.za/dataportal/index.php/catalog/NIDS/.

**ORCID iDs**
Benn Sartorius http://orcid.org/0000-0001-6761-2325
Pauline Scheelbeek http://orcid.org/0000-0002-6209-2284

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
