## [Reviewer comments · BMJ Open]

ARTICLE DETAILS

TITLE (PROVISIONAL)	Spatial-temporal trends and risk factors for under-nutrition and obesity among children (<5 years) in South Africa, 2008-2017: findings from a nationally representative longitudinal panel survey
AUTHORS	Sartorius, Benn; Sartorius, Kurt; Green, Rosemary; Lutge, Elizabeth; Scheelbeek, Pauline; Tanser, Frank; Dangour, Alan; Slotow, Rob

VERSION 1 – REVIEW

REVIEWER	Brandon Parkes Imperial College London, UK
REVIEW RETURNED	11-Nov-2019

GENERAL COMMENTS	P 4, Lines 19 & 25, neither of the references given for SA-NIDS dataset, which forms the basis of the paper, describe the dataset. Please include a reference that directly describes SA-NIDS: http://www.nids.uct.ac.za/ P5, lines 28-57. Data analysis, Space-time Bayesian modelling. This section does not provide a sufficient justification for the choice of this method of analysis. For example, the decision to use a Bayesian hierarchical model is often driven by a motivation to borrow information between related diseases to provide stability in prevalence rates where individual disease numbers are low due to small areas or rarity of the disease. However, there is no discussion of how closely related thinness/wasting is to obesity. (see also comments on results) p5, line 44 the word "prevalence's" should not have an apostrophe. p6, line 5. The authors should not refer to the menu options (svy: tab) in the stats software they employed, rather they should provide details of the actual statistical methods employed by the software. p6, line 49. There seems to be a unnecessary opening bracket here before the word "and". p7, line 11. The sentence would be clearer if it started "one district each..." instead of "one district respectively..." p7, line 13. The phrase "estimated stunting prevalence" is not the correct way to refer to the results of the space-time Bayesian modelling. Consider changing to "posterior median smoothed prevalence of stunting" or similar. This comment applies to all mentions of the results of the modelling, including the figure legends.
--

	p6-p8. This comment applies to the results section as a whole. There is insufficient presentation of the results of the Bayesian spatio-temporal analysis. When employing this modelling method, there are various assumptions made, and some of the likely effects of these assumptions can be examined using the outputs of the model beyond the posterior median smoother prevalence that is presented here. For example, consider contrasting the posterior median of the shared spatial component with the components that capture the un-shared spatial effects. Given the different spatial patterns of wasting/thinness and obesity (p7, lines 38-44), there may be something interesting to say about how appropriate it is to attempt to borrow information between these 'related' diseases. Additionally the posterior probabilities should be presented to give the readers an indication of the uncertainty in the smoothed results. P8, line 15. The word 'significantly' is used, presumably to indicate $p < 0.05$. The authors should consider not using this short hand for when presenting the results (see for example The American Statistical Association Statement on p-Values DOI 10.1080/00031305.2016.1154108). P8, line 59. 'tacking' is mis-spelt. Change to tackling. P12, line 3 to 9. The authors should attempt to quantify the effect of the missing/invalid weight/height measurements in wave 2. Perhaps use a sensitivity analysis and present the results in the supplemental material. P13, line 39. The authors state "the current and future health costs of malnutrition... cannot be overstated" and yet there is a whole section titled "Cost of malnutrition, policy and research needs" concerned with quantifying the costs of malnutrition. Consider rewording.
--	--

REVIEWER	Di Fang University of Arkansas
REVIEW RETURNED	24-Nov-2019

GENERAL COMMENTS	I believe this article is well written and well presented. The data set employed is a public available longitudinal data for children under 5 years old in South Africa. This study employed a spatial-temporal approach to analyze the prevalence of obesity and stunting. Given the growth rate of children at different stages, the study look at children under 2 years old and children above 2 years old separately. The results are explained well and clearly. I have only a few suggestions.  1. The authors claimed one of the contribution to be the model. A discussion of why a spatial-temporal model is superior than, for example a time series model, should be included. If possible, comparative results can be included to show the difference in estimates with and without considering the spatial nature. For this reason, an equation should be included as well. 2. The number of observations is small in some age categories. Even though sample weights are included a power analysis may be useful to convincing the readers of the validity of sample. 3. Figure a1) shows a wide, overlapping confidence intervals for all regions at all years. How does the authors arrive at statistical difference in time and space? Further tests may be needed.
---

	4. I understand this is a study of association. However, it seems like birth-weight and SES are really the driving factors, which is not surprising. The authors should discuss how these factors change over time as well as the policy implications. 5. I assume the authors used a spatial polygon to indicate relations in space. However, the construction of weight matrix should be explained in the text as well as the spatial test (e.g. Moran's I) to indicate the need of a spatial model. Thank you for the opportunity to review your work.
--	--

VERSION 1 – AUTHOR RESPONSE

Comment	Response
Reviewer: 1	Many thanks for the very useful and insightful comments.
P 4, Lines 19 & 25, neither of the references given for SA-NIDS dataset, which forms the basis of the paper, describe the dataset. Please include a reference that directly describes SA-NIDS: http://www.nids.uct.ac.za/	Agreed. The previous references 13 and 14 have been removed and replaced with following references that directly describes the methodology: 13. Leibbrandt M, Woolard I, de Villiers L. Methodology: Report on NIDS wave 1. Technical paper. 2009;1. 14. Southern Africa Labour and Development Research Unit. National Income Dynamics Study 2017, Wave 5 [dataset]. Version 1.0.0 In: Pretoria: Department of Planning M, and Evaluation [funding agency]. Cape Town: Southern Africa Labour and Development Research Unit [implementer], 2018. ed.: Cape Town: DataFirst [distributor], 2018., 2018. Secondly we have also added (under Methods, Data) the following full URL for where the underlying data can be accessed: http://www.nids.uct.ac.za/nids-data/data-access https://www.datafirst.uct.ac.za/dataportal/index.php/catalog/NIDS/
P5, lines 28-57. Data analysis, Space-time Bayesian modelling. This section does not provide a sufficient justification for the choice of this method of analysis. For example, the decision to use a Bayesian hierarchical model is often driven by a motivation to borrow information between related diseases to provide stability in prevalence rates where individual disease numbers are low due to small areas or rarity of the disease. However, there is no discussion of how closely related thinness/wasting is to	Agreed. The reviewer is correct. We have performed additional supplementary analyses (using GeoDa: Anselin L, Syabri I, Kho Y. GeoDa: an introduction to spatial data analysis. Geographical analysis. 2006 Jan;38(1):5-22) which assesses pairwise correlation/association between the 3 outcomes as well as bivariate Moran's I to assess if there was significant spatial autocorrelation between the outcomes. This analysis suggests that there is no significant association between stunting and thinness/wasting while there is weak positive but significant spatial autocorrelation between stunting and obesity prevalence as well as weak negative spatial correlation between thinness and obesity (please see detailed analyses below). These additional analyses can also be found in the revised supplementary material (Supplementary Section 1). Given this we have reformulated the joint Bayesian model to remove the shared spatial and temporal effects between 3 outcomes (please see Supplementary Section 2 for the revised model formulation). However, given that significant spatial heterogeneity were identified for all 3 outcomes using univariate Moran's I statistics (please see response below to reviewer 2), we have retained a Bayesian spatial-temporal formulation to model each of the outcomes independently.

obesity. (see also comments on results)

Based on the following review (Anderson C, Ryan LM. A Comparison of Spatio-Temporal Disease Mapping Approaches Including an Application to Ischaemic Heart Disease in New South Wales, Australia. *Int J Environ Res Public Health*. 2017 Feb 3;14(2):146. doi: 10.3390/ijerph14020146. PMID: 28165383; PMCID: PMC5334700.) we identified and fitted a spatial-temporal model for each outcome independently using the approach proposed by Martínez-Beneito MA, López-Quilez A, Botella-Rocamora P. An autoregressive approach to spatio-temporal disease mapping. *Statistics in medicine*. 2008 Jul 10;27(15):2874-89. The aforementioned paper “offers an autoregressive approach to spatio-temporal disease mapping by fusing ideas from autoregressive time series in order to link information in time and by spatial modelling to link information in space”. Furthermore, the authors concluded that an autoregressive model which only includes the spatial term for every period, leaving out the heterogeneous term resulted in a more parsimonious description of risk behaviour. We have also added additional text in the main methods section of the paper to better explain the rationale for the revised model formulation/approach (page 5). We have also included the following additional detail in the methods on page 7: “We used two-chain MCMC simulation for parameter estimation and Gelman-Rubin statistics/plots were used to assess model convergence/stability and where the Monte Carlo error for each parameter of interest was less than 5% of the sample standard deviation (Supplementary Material 3). For model validation, we firstly compared the observed and fitted prevalence values to assess overall model adequacy and fit and secondly, performed an out of sample validation using a random 10% sample with observed data. These analyses can be found in the Supplementary Material 4.”

```
. spearman stunted_svy thin_svy
```

```
Number of obs = 256  
Spearman's rho = 0.0729
```

```
Test of Ho: stunted_svy and thin_svy are independent  
Prob > |t| = 0.2452
```

```
. gllamm stunted_svy thin_svy, i(id)
```

```
number of level 1 units = 256  
number of level 2 units = 52
```

```
Condition Number = 14.594452
```

```
gllamm model
```

```
log likelihood = 283.93295
```

```
-----  
stunted_svy | Coef. Std. Err. z P>|z| [95% Conf. Interval]  
-----+-----  
thin_svy | .0385636 .0726234 0.53 0.595 -.1037757  
.1809028  
_cons | .1082981 .0061531 17.60 0.000 .0962381  
.120358  
-----
```

```
Variance at level 1  
-----
```

.00637033 (.00056306)

Variances and covariances of random effects

***level 2 (id)

var(1): 2.643e-24 (5.133e-14)

Bivariate Moran's I (using wave 5 as an example) suggests almost no spatial autocorrelation between stunting and thinness (Moran's I=-0.037, p=0.290)

permutations: 99999

pseudo p-value: 0.290100

t: -0.0372 E[I]: -0.0196 mean: 0.0007 sd: 0.0690 z-value: -0.5492

. spearman stunted_svy obese_svy

Number of obs = 256
Spearman's rho = 0.2051

Test of Ho: stunted_svy and obese_svy are independent
Prob > |t| = 0.0010

. gllamm stunted_svy obese_svy , i(id)

number of level 1 units = 256
number of level 2 units = 52

Condition Number = 10.565877

gllamm model

log likelihood = 292.58012

stunted_svy	Coef.	Std. Err.	z	P> z	[95% Conf. Interval]
					
obese_svy 	.1980684	.0475478	4.17	0.000	.1048765
.2912604					
_cons	.0791266	.0090305	8.76	0.000	.0614272
.0968261					

Variance at level 1

.00580379 (.00057983)

Variances and covariances of random effects

***level 2 (id)

var(1): .00015837 (.00029997)

permutations: 99999
pseudo p-value: 0.025380

t: -0.1350 E[t]: -0.0196 mean: 0.0001 sd: 0.0689 z-value: -1.9600

```
. spearman thin_svy obese_svy
```

Number of obs = 256

Spearman's rho = -0.1424

Test of Ho: thin_svy and obese_svy are independent

Prob > |t| = 0.0227

```
. gllamm thin_svy obese_svy , i(id)
```

number of level 1 units = 256

number of level 2 units = 52

Condition Number = 10.976401

gllamm model

log likelihood = 324.36079

thin_svy	Coef.	Std. Err.	z	P> z	[95% Conf. Interval]
obese_svy	-.067802	.040258	-1.68	0.092	-.1467062
.0111022					
_cons	.0602269	.0078037	7.72	0.000	.0449319
.0755218					

Variance at level 1

.00447574 (.00044278)

Variances and covariances of random effects

***|level 2 (id)

var(1): .00018259 (.00023176)

permutations: 99999
pseudo p-value: 0.020230

I: -0.1441 E[I]: -0.0196 mean: 0.0057 sd: 0.0710 z-value: -2.1119

With regards to the shared temporal effect this we think can be retained as all 3 outcomes appear to have a negative coefficient associated with increasing panel or wave

```
. gllamm stunted_svy year , i(id)
```

number of level 1 units = 256
number of level 2 units = 52

Condition Number = 31.724715

gllamm model

log likelihood = 293.64743

```
-----+-----  
stunted_svy |   Coef.   Std. Err.   z   P>|z|   [95% Conf. Interval]  
-----+-----  
      year | -0.0153423 .0033894 -4.53 0.000 -0.0219855 -  
.0086992  
      _cons | .1563577 .0112694 13.87 0.000 .1342702  
.1784453  
-----
```

Variance at level 1

```
-----  
.00590475 (.00052191)
```

Variances and covariances of random effects

```
-----  
***|level 2 (id)
```

```
var(1): 8.887e-19 (4.854e-11)  
-----
```

```
. gllamm thin_svy year , i(id)
```

number of level 1 units = 256

number of level 2 units = 52

Condition Number = 37.175479

gllamm model

log likelihood = 327.11892

```
-----+-----  
thin_svy |   Coef.   Std. Err.   z   P>|z|   [95% Conf. Interval]  
-----+-----  
      year | -0.0084373 .0028941 -2.92 0.004 -0.0141096 -  
.002765  
      _cons | .0749857 .0098979 7.58 0.000 .0555862  
.0943852  
-----
```

Variance at level 1

```
-----  
.00430301 (.00042507)
```

Variances and covariances of random effects

```
-----  
***|level 2 (id)
```

```
var(1): .00027197 (.0002388)  
-----
```

```
. gllamm obese_svy year , i(id)
```

	<pre> number of level 1 units = 256 number of level 2 units = 52 Condition Number = 21.597249 gllamm model log likelihood = 215.4003 ----- obese_svy Coef. Std. Err. z P> z [95% Conf. Interval] -----+----- year -0.0112194 .0043125 -2.60 0.009 -0.0196717 - .0027671 _cons .1905201 .0155017 12.29 0.000 .1601374 .2209029 ----- Variance at level 1 ----- .00954712 (.00094327) Variances and covariances of random effects ----- ***level 2 (id) var(1): .00175973 (.00074487) ----- </pre>
p5, line 44 the word "prevalence's" should not have an apostrophe.	Agreed. This has been corrected.
p6, line 5. The authors should not refer to the menu options (svy: tab) in the stats software they employed, rather they should provide details of the actual statistical methods employed by the software.	Agreed. We have removed reference to the Stata menu options and provided more details regarding the statistical methods employed. Please see page 5.
p6, line 49. There seems to be a unnecessary opening bracket here before the word "and".	Agreed. This has been removed.
p7, line 11. The sentence would be clearer if it started "one district each..." instead of "one district respectively..."	Agreed. This has been corrected to "one district each..." on page 7.
p7, line 13. The phrase "estimated stunting prevalence" is not the correct way to refer to the results of the space-time Bayesian modelling. Consider changing to	Agreed. Relevant sentences (on page 7) have been revised as suggested as well as figure legends (page 8).

"posterior median smoothed prevalence of stunting" or similar. This comment applies to all mentions of the results of the modelling, including the figure legends.	
p6-p8. This comment applies to the results section as a whole. There is insufficient presentation of the results of the Bayesian spatio-temporal analysis. When employing this modelling method, there are various assumptions made, and some of the likely effects of these assumptions can be examined using the outputs of the model beyond the posterior median smoother prevalence that is presented here. For example, consider contrasting the posterior median of the shared spatial component with the components that capture the un-shared spatial effects. Given the different spatial patterns of wasting/thinness and obesity (p7, lines 38-44), there may be something interesting to say about how appropriate it is to attempt to borrow information between these 'related' diseases. Additionally, the posterior probabilities should be presented to give the readers an indication of the uncertainty in the smoothed results.	Agreed. We have performed additional supplementary analyses (using GeoDa: Anselin L, Syabri I, Kho Y. GeoDa: an introduction to spatial data analysis. Geographical analysis. 2006 Jan;38(1):5-22) which assesses pairwise correlation/association between the 3 outcomes as well as bivariate Moran's I to assess if there was significant spatial autocorrelation between the outcomes. This analysis suggests that there is no significant association between stunting and thinness/wasting while there is weak positive but significant spatial autocorrelation between stunting and obesity prevalence as well as weak negative spatial correlation between thinness and obesity (please see detailed analyses below). These additional analyses can also be found in the revised supplementary material (Supplementary Section 1). Given this we have reformulated the joint Bayesian model to remove the shared spatial and temporal effects between 3 outcomes (please see Supplementary Section 2 for the revised model formulation). However, given that significant spatial heterogeneity were identified for all 3 outcomes using univariate Moran's I statistics (please see response below to reviewer 2), we have retained a Bayesian spatio-temporal formulation to model each of the outcomes independently. Based on the following review (Anderson C, Ryan LM. A Comparison of Spatio-Temporal Disease Mapping Approaches Including an Application to Ischaemic Heart Disease in New South Wales, Australia. Int J Environ Res Public Health. 2017 Feb 3;14(2):146. doi: 10.3390/ijerph14020146. PMID: 28165383; PMCID: PMC5334700.) we identified and fitted a spatial-temporal model for each outcome independently using the approach proposed by Martínez-Beneito MA, López-Quilez A, Botella-Rocamora P. An autoregressive approach to spatio-temporal disease mapping. Statistics in medicine. 2008 Jul 10;27(15):2874-89. The aforementioned paper "offers an autoregressive approach to spatio-temporal disease mapping by fusing ideas from autoregressive time series in order to link information in time and by spatial modelling to link information in space". Furthermore, the authors concluded that an autoregressive model which only includes the spatial term for every period, leaving out the heterogeneous term resulted in a more parsimonious description of risk behaviour. We have also added additional text in the main methods section of the paper to better explain the rationale for the revised model formulation/approach (pages 5-6). We have also included the following additional detail in the methods on page 7: "We used two-chain MCMC simulation for parameter estimation and Gelman-Rubin statistics/plots were used to assess model convergence/stability and where the Monte Carlo error for each parameter of interest was less than 5% of the sample standard deviation (Supplementary Material 3). For model validation, we firstly compared the observed and fitted prevalence values to assess overall model adequacy and fit and secondly, performed an out of sample validation using a random 10% sample with observed data. These analyses can be found in the Supplementary Material 4." Lastly we have now included additional visualisations for the width of the uncertainty intervals for the posterior estimates to given a clearer indication of the uncertainty in the smoothed results (Supplementary

Section 7). Furthermore, we also calculated exceedance probabilities for the 5% target threshold for thinness/wasting based on international target thresholds to more conclusively highlight districts which still exceeded this level in wave 5. Sample of the table from Supplementary Section 7 as an example:

District	wave	stunting	95% UI		thinness	95% UI	
Mangaung(MAN)	1	0.376	0.2267	0.5446	0.08848	0.01452	0.2
Nelson Mandela Bay(NMA)	1	0.1189	0.04999	0.2169	0.04979	0.01135	0.1
City of Tshwane(TSH)	1	0.1683	0.1008	0.254	0.1242	0.06282	0.2
City of Johannesburg(JHB)	1	0.1089	0.05726	0.1785	0.05937	0.02058	0.1
Buffalo City(BUF)	1	0.3057	0.1065	0.5683	0.2221	0.04924	0.5
City of Cape Town(CPT)	1	0.08183	0.03667	0.1476	0.08437	0.01581	0.2
West Coast(DC1)	1	0.1033	0.04348	0.1936	0.09203	0.0153	0.2
Cacadu(DC10)	1	0.2199	0.1344	0.3257	0.08308	0.02311	0.2
Amathole(DC12)	1	0.2096	0.099	0.3623	0.1707	0.0536	0.3

As there are no specific target thresholds for stunting/obesity against which we can calculate exceedance probabilities we have not estimated any particular exceedance p-value for these.

P8, line 15. The word 'significantly' is used, presumably to indicate $p < 0.05$. The authors should consider not using this short hand for when presenting the results (see for example The American Statistical Association Statement on p-Values DOI 10.1080/00031305.2016.1154108)

Agreed. We have toned down use of “significance” vs “non-significance” in the results narrative and interpretation. Please see revised results narrative on pages 6-8. Furthermore, it should be noted that Table 1 we presented 95% confidence intervals around prevalence point estimates rather than p-values i.e. “emphasize estimation over testing”. We have now also modified Table 2 to also include 95% confidence intervals around prevalence point estimates for this reason.

P8, line 59. 'tacking' is misspelt. Change to tackling.

Thanks. This has been corrected (now on page 9).

P12, line 3 to 9. The authors should attempt to quantify the effect of the missing/invalid weight/height measurements in wave 2. Perhaps use a sensitivity analysis and present the results in the supplemental material.

Agreed. We have now also performed a sensitivity analysis comparing the various socio-demographic characteristics by missing weight and height status. This has been included in the Supplementary Material (#5). Many of these characteristics were not significantly different by missing weight/height status which strengthens the argument that these were potentially missing at random.
 “Summary: A comparison of missing weight/height proportions by various socio-demographic variables suggests that many were likely missing at random. Distributions of race, gender, household income, low birthweight, food security status, mother education category and father education category were not significantly different when comparing children with missing weight/height measurements to those with a valid weight/height measurement (please see analysis output below). However, age did significantly differ by missing status in that infants (<1 year of age) were significantly more likely to have a missing weight/height measurement compared to children aged 1-4 years. There also appeared to be significant differences in missing weight/height status by province of residence i.e. children in Mpumalanga, Western Cape for example had higher proportions of missing weight/height measurements among children under 5 ($p < 0.001$). Furthermore, missing weight/height measurements for

children were more significantly more likely among those children with younger mothers (<25 years of age).”
 E.g. race, gender, age, household income quantile. Please see supplementary material 5 for further comparisons.

. svy: tab race_ missing_height_weight, row ci
 (running tabulate on estimation sample)

Number of strata = 53 Number of obs = 16,649
 Number of PSUs = 1,076 Population size = 25,331,414
 Design df = 1,023

race_	missing_height_weight		Total
	0	1	
African	.8129	.1871	1
	[.8006,.8246]	[.1754,.1994]	
Coloured	.7803	.2197	1
	[.7437,.8129]	[.1871,.2563]	
Asian/In	.7593	.2407	1
	[.5708,.882]	[.118,.4292]	
White	.74	.26	1
	[.643,.8182]	[.1818,.357]	
Total	.8066	.1934	1
	[.7945,.8181]	[.1819,.2055]	

Key: row proportion
 [95% confidence interval for row proportion]

Pearson:
 Uncorrected chi2(3) = 32.5162
 Design-based F(2.49, 2551.53)= 1.7810 P = 0.1588

. svy: tab gender_ missing_height_weight, row ci
 (running tabulate on estimation sample)

Number of strata = 53 Number of obs = 19,138
 Number of PSUs = 1,218 Population size = 28,354,881
 Design df = 1,165

gender_	missing_height_weight		Total
	0	1	
Male	.8065	.1935	1
	[.7926,.8196]	[.1804,.2074]	
Female	.8102	.1898	1
	[.7951,.8245]	[.1755,.2049]	
Total	.8083	.1917	1
	[.7972,.819]	[.181,.2028]	

Key: row proportion
 [95% confidence interval for row proportion]

Pearson:
 Uncorrected chi2(1) = 0.4400
 Design-based F(1, 1165) = 0.1697 P = 0.6805

. svy: tab age_ missing_height_weight, row ci
 (running tabulate on estimation sample)

Number of strata = 53 Number of obs = 19,201
 Number of PSUs = 1,227 Population size = 28,456,616
 Design df = 1,174

age_	missing_height_weight		Total
	0	1	
0	.4596	.5404	1
	[.4362,.4832]	[.5168,.5638]	
1	.8581	.1419	1
	[.8308,.8816]	[.1184,.1692]	
2	.8764	.1236	1
	[.8573,.8933]	[.1067,.1427]	
3	.8952	.1048	1
	[.8726,.9142]	[.0858,.1274]	
4	.9015	.0985	1
	[.8847,.916]	[.084,.1153]	
Total	.8083	.1917	1
	[.7972,.8189]	[.1811,.2028]	

Key: row proportion
 [95% confidence interval for row proportion]

Pearson:
 Uncorrected chi2(4) = 3267.7805
 Design-based F(3.41, 3999.27)= 238.9174 **P = 0.0000**

. svy: tab hh_inc missing_height_weight, row ci
 (running tabulate on estimation sample)

Number of strata = 53 Number of obs = 18,289
 Number of PSUs = 1,195 Population size = 26,887,499
 Design df = 1,142

hh_inc	missing_height_weight		Total
	0	1	
1	.8032	.1968	1
	[.7792,.8251]	[.1749,.2208]	
2	.8286	.1714	1

	<pre> [.8012,.853] [.147,.1988] 3 .8289 .1711 1 [.8084,.8475] [.1525,.1916] 4 .8076 .1924 1 [.7751,.8365] [.1635,.2249] 5 .7862 .2138 1 [.7578,.812] [.188,.2422] Total .8096 .1904 1 [.7982,.8205] [.1795,.2018] ----- Key: row proportion [95% confidence interval for row proportion] Pearson: Uncorrected chi2(4) = 32.2620 Design-based F(3.67, 4186.36)= 1.9756 P = 0.1017 We have included an additional sentence in the methods detailing this additional sensitivity analysis. Furthermore, we have also added an additional statement to the limitations in this regard. </pre>
P13, line 39. The authors state "the current and future health costs of malnutrition... cannot be overstated" and yet there is a whole section titled "Cost of malnutrition, policy and research needs" concerned with quantifying the costs of malnutrition. Consider rewording.	Agreed. We have clarified this statement in the concluding section as follows: "The current and future health cost of malnutrition among South African children is likely substantial based on previous costing estimates ".
Reviewer: 2	
I believe this article is well written and well presented. The data set employed is a public available longitudinal data for children under 5 years old in South Africa. This study employed a spatial-temporal approach to analyze the prevalence of obesity and stunting. Given the growth rate of children at different stages, the study look at children under 2 years old and children above 2 years old separately. The results are explained well and clearly. I have only a few suggestions.	Many thanks. The comments received below were very useful and insightful. The authors have attempted to address these in detail.
1. The authors claimed one of the contribution to be the model. A discussion of why	Agreed. We have now also included results from spatial autocorrelation tests (using GeoDa: Anselin L, Syabri I, Kho Y. GeoDa: an introduction to spatial data analysis. Geographical

a spatial-temporal model is superior than, for example a time series model, should be included. If possible, comparative results can be included to show the difference in estimates with and without considering the spatial nature. For this reason, an equation should be included as well.

analysis. 2006 Jan;38(1):5-22) in the supplementary section (please see below) which justify the choice of a spatial model i.e. Moran's I tests suggest moderate/high significant autocorrelation for all 3 anthropometric classifications (please see detailed output below). We have also included additional details in the methods section of the paper detailing this.

Please see revised methods, page 5: "We assessed for the presence of univariate and bivariate spatial autocorrelation for the three anthropometric classifications using Moran's I statistics. This analysis was performed using GeoDa. Based on these tests it appeared that there was no prominent bivariate spatial autocorrelation between the three measures but that each measure was significant heterogeneous across space to warrant the use of a spatial model (supplementary section 1)." We also performed additional supplementary analyses which bivariate Moran's I to assess if there was significant spatial autocorrelation between the outcomes. This analysis suggests that there is no significant association between stunting and thinness/wasting while there is weak positive but significant spatial autocorrelation between stunting and obesity prevalence as well as weak negative spatial correlation between thinness and obesity (please see detailed analyses below). These additional analyses can also be found in the revised supplementary material (Supplementary Section 1).

Given this we have reformulated the joint Bayesian model to remove the shared spatial and temporal effects between 3 outcomes (please see Supplementary Section 2 for the revised model formulation). However, given that significant spatial heterogeneity were identified for all 3 outcomes using univariate Moran's I statistics (please see response below to reviewer 2), we have retained a Bayesian spatial-temporal formulation to model each of the outcomes independently. Based on the following review (Anderson C, Ryan LM. A Comparison of Spatio-Temporal Disease Mapping Approaches Including an Application to Ischaemic Heart Disease in New South Wales, Australia. *Int J Environ Res Public Health*. 2017 Feb 3;14(2):146. doi: 10.3390/ijerph14020146. PMID: 28165383; PMCID: PMC5334700.) we identified and fitted a spatial-temporal model for each outcome independently using the approach proposed by Martínez-Beneito MA, López-Quilez A, Botella-Rocamora P. An autoregressive approach to spatio-temporal disease mapping. *Statistics in medicine*. 2008 Jul 10;27(15):2874-89. The aforementioned paper "offers an autoregressive approach to spatio-temporal disease mapping by fusing ideas from autoregressive time series in order to link information in time and by spatial modelling to link information in space". Furthermore, the authors concluded that an autoregressive model which only includes the spatial term for every period, leaving out the heterogeneous term resulted in a more parsimonious description of risk behaviour. We have also added additional text in the main methods section of the paper to better explain the rationale for the revised model formulation/approach (pages 5-6).

Please also note that the full equation for the space-time model formulation has been included in the revised methods on pages 5-7, namely:

"We employed Bayesian spatial-temporal modelling approach in an attempt to stabilise estimates at district level given that the primary sampling design was not developed to provide point estimates at this level of geographic disaggregation and resultant zero prevalence

estimates for particular districts and waves. We choose a Bayesian spatial-temporal formulation to model each of the anthropometric outcomes independently using an autoregressive approach, suggested by a recent methodological comparison, which fuses ideas from autoregressive time series to link information in time and by spatial modelling to link information in space. We also opted for an autoregressive model which only included the spatial term for every period and did not include a heterogeneous term which resulted in a more parsimonious description of risk.

Let Y_{ij} be the number of stunted, thin or obese children for the i th area and j th period, $i = 1, \dots, I$, $j = 1, \dots, J$, and n_{ij} the total number of children sampled in a given area and period. We assumed that Y_{ij} follows a binomial distribution i.e. $Y_{ij} \sim \text{binomial}(n_{ij}, \pi_{ij})$, $i = 1, \dots, 53$, $j = 1, \dots, 5$, where π it is the risk (prevalence) of stunting, thinness or obesity in region i in period j . As per Martinez-Beneito et al. we define the logit of the prevalence for a given anthropometric outcome for the first wave (or period) as the sum of an intercept and two random effects, namely:

$$\pi_{i1} = \mu + \alpha_1 + (1 - \rho^2)^{-1/2} \cdot (\theta_{i1} + \phi_{i1}), \quad i = 1, \dots, I$$

$$\theta_{i1} \sim \text{Normal}(0, \sigma^2_\theta), \quad i = 1, \dots, I$$

$$\phi_1 = (\phi_{11}, \dots, \phi_{I1}) \sim \text{CAR.normal}(\sigma^2_\phi)$$

and subsequent time periods $2, \dots, J$ as:

$$\pi_{ij} = \mu + \alpha_j + \rho \cdot (\pi_{i(j-1)} - \mu - \alpha_{j-1}) + \theta_{ij} + \phi_{ij}, \quad \text{for } i = 1, \dots, I \text{ and } j = 2, \dots, J$$

$$\theta_{ij} \sim \text{Normal}(0, \sigma^2_\theta), \quad \text{for } i = 1, \dots, I \text{ and } j = 2, \dots, J$$

$$\phi_j \sim \text{CAR.normal}(\sigma^2_\phi), \quad \text{for } j = 2, \dots, J$$

$$\alpha = (\alpha_1, \alpha_2, \dots, \alpha_J) \sim \text{CAR.normal}(\sigma^2_\alpha)$$

where ϕ , the spatial random effect, assumes an intrinsic Gaussian conditionally autoregressive distribution (abbreviated above as CAR.normal), whereby the spatially correlated random effect of the i th region (ϕ_i) is based on the sum of its weighted neighbourhood values. We used an adjacency matrix of common boundaries (neighbours) of a given region when modelling this parameter. The heterogeneous or unstructured random effect is represented by θ and is included to ensure sufficient flexibility for estimates in close regions that is not captured by the spatially structured term. The spatial and heterogeneous random effect terms are both independent in time and mutually independent in every period. Furthermore, ρ corresponds to the temporal correlation term, μ models the mean level of risks for all the periods and regions and α_1 models the mean deviation of the risks in the first period from the mean level for all of them. A first-order random walk CAR.normal was also used as prior distribution for α .

The following prior distributions were assumed for the parameters defined above:

$$\sigma^2_\theta, \sigma^2_\phi, \sigma^2_\alpha \sim \text{Gamma}(0.5, 0.0005)$$

$$\rho \sim \text{Uniform}(-1, 1)$$

	$\mu \sim \text{Normal}(0,c)$ The prior distribution on the temporal correlation parameter (ρ) was chosen to ensure the stationarity of the time series, considering that it has an order 1 autoregressive structure. We chose inverse gamma distributions for the variance parameters with values of 0.5 and 0.0005 as suggested by Wakefield et al” We have also included the following additional detail in the methods on page 7: “We used two-chain MCMC simulation for parameter estimation and Gelman-Rubin statistics/plots were used to assess model convergence/stability and where the Monte Carlo error for each parameter of interest was less than 5% of the sample standard deviation (Supplementary Material 3). For model validation, we firstly compared the observed and fitted prevalence values to assess overall model adequacy and fit and secondly, performed an out of sample validation using a random 10% sample with observed data. These analyses can be found in the Supplementary Material 4.”																					
2. The number of observations is small in some age categories. Even though sample weights are included a power analysis may be useful to convincing the readers of the validity of sample.	Agreed. We performed a post hoc power analysis to assess the minimum effect size detectable among infants which has the smallest number of observations. The post hoc power analysis suggests that the sample size in the smallest age group has the power to detect a small effect size ($w \sim 0.1$ based on Cohens rules of thumb [Cohen, 1988]) when using a chi-square test with 2x9 cells (maximum number of cells tested in our analyses i.e. binary nutritional classification versus province of residence) with 80% power and 5% alpha or type I error. χ^2 tests - Goodness-of-fit tests: Contingency tables Analysis: Post hoc: Compute achieved power Input:  Effect size w=0.11 α err prob=0.05 Total sample size=1277 Df=8  Output:  Noncentrality parameter λ=15.4517000 Critical χ^2=15.5073131 Power (1-β err prob)=0.8133607  Cohen, J (1988) Statistical power analysis for the behavioral sciences (2nd ed.). Hillsdale, NJ: Erlbaum. Please see following table in Kotrlik, JW and Williams, HA (2003) The incorporation of effect size in information technology, learning, and performance research. Information Technology, Learning, and Performance Journal 21(1) 1-7.	Effect size w	=	0.11	α err prob	=	0.05	Total sample size	=	1277	Df	=	8	Noncentrality parameter λ	=	15.4517000	Critical χ^2	=	15.5073131	Power (1- β err prob)	=	0.8133607
Effect size w	=	0.11																				
α err prob	=	0.05																				
Total sample size	=	1277																				
Df	=	8																				
Noncentrality parameter λ	=	15.4517000																				
Critical χ^2	=	15.5073131																				
Power (1- β err prob)	=	0.8133607																				

Effect Size	Use	Small	Medium	Large
Correlation inc Phi		0.1	0.3	0.5
Cramer's V	r x c frequency tables	0.1	0.3	0.5
Difference in arcsines	Comparing two proportions	0.2	0.5	0.8
η^2	Anova	0.01	0.06	0.14
omega-squared	Anova; See Field (2013)	0.01	0.06	0.14
Multivariate eta-squared	one-way MANOVA	0.01	0.06	0.14
Cohen's f	one-way an(c)ova (regression)	0.1	0.25	0.4
η^2	Multiple regression	0.02	0.13	0.26
κ^2	Mediation analysis	0.01	0.09	0.25
Cohen's f	Multiple Regression	0.14	0.39	0.59
Cohen's d	t-tests	0.2	0.5	0.8
Cohen's ω	chi-square	0.1	0.3	0.5
Odds Ratios	2 by 2 tables	1.5	3.5	9
Odds Ratios	p vs 0.5	0.55	0.65	0.75
Average Spearman rho	Friedman test	0.1	0.3	0.5

We have included the above in the supplementary material (section 8).

3. Figure a1) shows a wide, overlapping confidence intervals for all regions at all years. How does the authors arrive at statistical difference in time and space? Further tests may be needed.

Agreed. Reviewer 1 also raised concerns regarding the use of the terminology "significance" as pertaining to p-values. We have removed reference to the word significance. The reviewer is also correct that most pairwise differences referred to in Figure 1a-c are not statistically significant given the overlapping uncertainty intervals. For example

```
. svy: tab stunted_final province if year==1, col
(running tabulate on estimation sample)
```

```
Number of strata = 52          Number of obs = 2,079
Number of PSUs = 345         Population size =
3,248,532
Design df = 293
```

```
-----+-----
stunted_f |                province
inal | Eastern Free Sta Gauteng KwaZulu- Limpopo
Mpumalan North We Northern Western Total
-----+-----
0 | .8277 .892 .8654 .9348 .8602 .9002 .9077
.8682 .9395 .8901
1 | .1723 .108 .1346 .0652 .1398 .0998 .0923
.1318 .0605 .1099
|
Total | 1 1 1 1 1 1 1
1 1
```

Key: column proportion

Pearson:
Uncorrected chi2(8) = 30.8524
Design-based F(6.14, 1798.23)= 2.0935 P = 0.0496

```
. svy: tab stunted_final province if year==2, col
```

(running tabulate on estimation sample)

Number of strata = 52 Number of obs = 1,499
 Number of PSUs = 372 Population size = 2,410,873
 Design df = 320

stunted_f		province						
inal		Eastern	Free Sta	Gauteng	KwaZulu-	Limpopo	Total	
Mpumalan	North We	Northern	Western					
0		.8224	.8575	.8581	.8492	.8028	.9138	.8207
.9166		.8378	.844					
1		.1776	.1425	.1419	.1508	.1972	.0862	.1793
.0834		.1622	.156					
Total		1	1	1	1	1	1	1
1	1							

Key: column proportion

Pearson:

Uncorrected chi2(8) = 7.8869

Design-based F(4.89, 1565.26)= 0.4326 P = 0.8222

Note: Strata with single sampling unit centered at overall mean.

. svy: tab stunted_final province if year==3, col
 (running tabulate on estimation sample)

Number of strata = 52 Number of obs = 2,916
 Number of PSUs = 583 Population size = 4,526,869
 Design df = 531

stunted_f		province						
inal		Eastern	Free Sta	Gauteng	KwaZulu-	Limpopo	Total	
Mpumalan	North We	Northern	Western					
0		.8562	.811	.892	.8712	.8449	.8247	.8517
.9194		.9149	.8683					
1		.1438	.189	.108	.1288	.1551	.1753	.1483
.0806		.0851	.1317					
Total		1	1	1	1	1	1	1
1	1							

Key: column proportion

Pearson:

Uncorrected chi2(8) = 22.3351

Design-based F(6.79, 3603.59)= 1.0068 P = 0.4231

. svy: tab stunted_final province if year==5, col
(running tabulate on estimation sample)

Number of strata = 52 Number of obs = 3,740
 Number of PSUs = 801 Population size = 4,725,873
 Design df = 749

stunted_f		province						
inal		Eastern	Free Sta	Gauteng	KwaZulu-	Limpopo	Total	
Mpumalan		North We	Northern	Western				
0	.9285	.8677	.9524	.9314	.907	.912	.9066	
.8957	.9118	.9244						
1	.0715	.1323	.0476	.0686	.093	.088	.0934	
.1043	.0882	.0756						
Total	1	1	1	1	1	1	1	
1	1							

Key: column proportion

Pearson:
 Uncorrected chi2(8) = 23.9878
 Design-based F(5.42, 4062.34) = 1.2048 P = 0.3027

Etc.

4. I understand this is a study of association. However, it seems like birth-weight and SES are really the driving factors, which is not surprising. The authors should discuss how these factors change over time as well as the policy implications.

Agreed. An analysis of the change in these factors over time suggests that birthweight and related LBW/HBW classifications are not changing substantially over this period:

However, household income (SES) appears to have increased fairly rapidly over this period:

We have included substantial discussion around the association of birthweight and income (SES) with under and over nutrition in relation to your findings. Please see lines 296-317 and 335-342. Additional reference to rapidly increasing income in this context and policy related implications have been added to the discussion. We have expanded the discussion around the policy related implications of birthweight and SES under section “Cost of malnutrition, policy and research needs”. Lines 397-399: “A further contextual trend which may further compound this problem is the rapidly rising median household income observed over the period (from ZAR1400 in 2008 to ZAR 3640 by 2017).”

5. I assume the authors used a spatial polygon to indicate relations in space. However, the construction of weight matrix should be explained in the text as well as the spatial test (e.g. Moran's I) to indicate the need of a spatial model.

Agreed. We have now included spatial autocorrelation tests in the supplementary section (please see below) which justify the choice of a spatial model i.e. Moran's I tests suggest moderate/high significant autocorrelation for all 3 anthropometric classifications (please see detailed output below and also in the revised supplementary material).

permutations: 99999
pseudo p-value: 0.000020

t: 0.4027 E[I]: -0.0196 mean: -0.0202 sd: 0.0923 z-value: 4.5834

Moran's I: 0.371

permutations: 99999
pseudo p-value: 0.000130

t: 0.3707 E[I]: -0.0196 mean: -0.0196 sd: 0.0916 z-value: 4.2602

permutations: 99999
 pseudo p-value: 0.000010

I: 0.5600 E[I]: -0.0196 mean: -0.0198 sd: 0.0918 z-value: 6.3149

We have also included additional details in the methods section of the paper detailing this as well as the weight matrix used in the autoregressive spatial-temporal model. Please see page 6: “ ϕ , the spatial random effect, assumes an intrinsic Gaussian conditionally autoregressive distribution (abbreviated above as CAR.normal), whereby the spatially correlated random effect of the i^{th} region (ϕ_i) is based on the sum of its weighted neighbourhood values. We used an adjacency matrix of common boundaries (neighbours) of a given region when modelling this parameter.”

Thank you for the opportunity to review your work.

Many thanks for the useful comments which have improved the quality of this work.

FORMATTING AMENDMENTS (if any)

- The spelling of the name of co-author ‘Scheekbeek, Pauline’ in ScholarOne system is different from the main document ‘Scheelbeek P’. Kindly amend accordingly.

Scheekbeek is misspelt on the ScholarOne system. This authors surname should read “Scheelbeek” as currently listed in the manuscript.

VERSION 2 – REVIEW

REVIEWER	Brandon Parkes Imperial College London, UK
REVIEW RETURNED	02-Jan-2020

GENERAL COMMENTS	P5, line 125 please change 'significant' to 'significantly' P8&9 Results section. I do not feel my previous comment on the results section has been addressed: "This comment applies to the results section as a whole. There is insufficient presentation of the results of the Bayesian spatio-temporal analysis. When employing this modelling method, there are various assumptions made, and some of the likely effects of these assumptions can be examined using the outputs of the model beyond the posterior median smoother prevalence that is presented here. For example, consider contrasting the posterior median of the shared spatial component with the components that capture the un-shared spatial effects. Given the different spatial patterns of wasting/thinness and obesity (p7, lines 38-44), there may be something interesting to say about how appropriate it is to attempt to borrow information between these 'related' diseases. Additionally the posterior probabilities should be presented to give the readers an indication of the uncertainty in the smoothed results."
--

REVIEWER	Di Fang University of Arkansas
REVIEW RETURNED	31-Jan-2020

GENERAL COMMENTS	The authors have adequately addressed my previous comments.
---

VERSION 2 – AUTHOR RESPONSE

Reviewer 1

P5, line 125 please change 'significant' to 'significantly'

Response: Agreed, we have changed this to “significantly” as suggested.

P8&9 Results section. I do not feel my previous comment on the results section has been addressed: "This comment applies to the results section as a whole. There is insufficient presentation of the results of the Bayesian spatio-temporal analysis. When employing this modelling method, there are various assumptions made, and some of the likely effects of these assumptions can be examined using the outputs of the model beyond the posterior median smoother prevalence that is presented here.

Response: Agreed. We have included the following the methods, page 6, lines 162-163:

“Sensitivity of the estimates to prior specification was assessed by repeating the analysis with different hyper parameters (Supplementary 4).”

The following additional analysis has also been added to supplementary 4 (please see section b):

b)

We concluded an additional sensitivity analysis to confirm whether the choice of hyper parameter may have affected the prevalence estimates. For the variance parameters, namely σ^2_v , σ^2_ϕ , σ^2_γ we assumed Gamma(0.5,0.0005) distributions as recommended by Wakefield (Wakefield J, Best N, Waller L. Bayesian approaches to disease mapping. Spatial epidemiology: methods and applications 2000:104-07.) for the Bayesian prevalence/exceedance probability estimates presented in the main text. We also tested whether changes to this prior may have affected the estimates. Other choices for this prior (Lawson A, Browne W, Vidal Rodeiro C. Disease Mapping with WinBUGS and MLWin. Chichester: John Wiley & Sons; 2003) that are commonly used include.

Gamma (0.001, 0.001)

Gamma (0.01,0.01)

Pairwise scatterplots of the posterior prevalence for the various gamma distribution choices for the hyper parameters below suggest that the model estimates were largely insensitive to the choice of distribution assumed:

For example, consider contrasting the posterior median of the shared spatial component with the components that capture the un-shared spatial effects. Given the different spatial patterns of wasting/thinness and obesity (p7, lines 38-44), there may be something interesting to say about how appropriate it is to attempt to borrow information between these 'related' diseases.

Response: Agreed. However, in response to your and reviewer 2's previous comments, we decided from further exploratory analyses that a joint spatial model was not appropriate as the

degree of bivariate spatial autocorrelation across the 3 nutritional classifications was not sufficient to warrant a shared component model. We have therefore implemented separate space-time autoregressive models for each outcome. Hence we cannot contrast the posterior median of the shared spatial component with the components that capture the un-shared spatial effects as there is no longer a shared component in the model parameterisation.

We have however included additional model related summaries (e.g. posteriors for the space - time random effects (phi and gamma respectively) as well as the unstructured space-time interaction term (nu)– please new supplementary 4) as suggested by the reviewer to provide more information for the interested reader to assess the overall model adequacy.

Supplementary 4: Model random effects posteriors

Spatial random effects (phi)
Stunting

Thinness/wasting

Obesity

Unstructured effects (2017) (nu)

Stunting

Thinness/wasting

Obesity

Temporal random effects (gamma)

The overall fit for each outcome model is presented in supplementary 6 (please see DIC statistics).

Please note we have included in the previous response to comments, detailed supplementary analyses related to the observed versus fitted values, out of sample validation as well as sensitivity analyses for the missing weight/height measurements.

Additionally, the posterior probabilities should be presented to give the readers an indication of the uncertainty in the smoothed results."

Response: We had included these additional analyses with uncertainty intervals and exceedance probabilities in the supplementary material (#9). As only wasting has a defined absolute target threshold of under 5%, we had thus previously estimated the exceedance probability for this threshold by district and year which is presented below. As per the 2025 global nutritional targets for stunting and obesity, namely 40% relative reduction in stunting from 2012 to 2025 and no increase in overweight/obesity from 2012 to 2025 respectively, we have now also included exceedance probability parameters for stunting and obesity as per these targets. The following exceedance parameters in the WINBUGS model are now parameterised as follows:

Stunting: $\text{exceedance1}[5,i] < -\text{step}((1-p1[5,i]/p1[3,i]) - 0.17)$ #17% is target reduction by 2017 from 2015
assuming target 40% reduction by 2025

Wasting: $\text{exceedance2}[j,i] < -\text{step}(p2[j,i] - 0.05)$ # reduce and maintain wasting to <5%

Obesity: $\text{exceedance3}[5,i] < -\text{step}(p3[5,i]/p3[3,i] - 1)$ # no increase in obesity from 2012 to 2017

In addition to the posterior prevalence and exceedance probabilities for the 3 nutritional classification presented in supplementary 9, we have also include additional results narrative in the main text which speaks to trends in space-time (at district and survey round level) which speak to the progress towards 2025 targets based on our estimated exceedance probabilities pertaining to these thresholds over the observed period of observation, namely 2008 to 2017. We have also included additional narrative results text on page 8/9 which speaks to these exceedance probabilities related to the WHO 2025 nutritional targets.

Supplementary 9: Full posterior prevalence estimates with 95% Bayesian uncertainty intervals (UIs) by district and year. Also includes exceedance probabilities for 17% reduction in stunting from wave 3 (2012) to wave 5 (2017) – to achieve 40% reduction from 2012 to 2025, 5% target threshold for wasting prevalence and no increase in obesity from wave 3 (2012) to wave 5 (2017) as per 2025 nutritional targets.

Province	District	wave	stunting	95% UI		Exceedance probability 17% reduction from wave 3	thinness	95% UI		Exceedance probability 5% target threshold
Eastern	Alfred Nzo(DC44)	1	9.2%	4.4%	16.1%	N/A	6.1%	2.3%	12.1%	0.6%
KwaZulu-	Amajuba(DC25)	1	9.7%	5.1%	15.7%	N/A	5.1%	2.1%	9.8%	0.4%
Eastern	Amathole(DC12)	1	14.8%	6.4%	27.4%	N/A	12.4%	3.9%	26.8%	0.9%
North West	Bojanala(DC37)	1	10.2%	4.6%	18.4%	N/A	5.7%	1.9%	12.3%	0.5%
Eastern	Buffalo City(BUF)	1	19.0%	8.3%	35.1%	N/A	14.2%	4.0%	33.4%	0.9%
Eastern	Cacadu(DC10)	1	21.7%	12.9%	32.5%	N/A	8.0%	2.2%	19.8%	0.7%
Western	Cape Winelands(DC2	1	12.5%	4.7%	25.8%	N/A	9.7%	4.4%	17.1%	0.9%
Limpopo	Capricorn(DC35)	1	12.4%	6.5%	20.6%	N/A	10.1%	4.6%	18.2%	0.9%
Western	Central Karoo(DC5)	1	16.0%	9.0%	24.9%	N/A	7.6%	3.2%	14.3%	0.8%
Eastern	Chris Hani(DC13)	1	9.7%	4.7%	17.0%	N/A	7.4%	2.1%	18.3%	0.6%
Western	City of Cape Town	1	8.1%	4.0%	13.8%	N/A	9.0%	2.4%	21.6%	0.7%
Gauteng	City of Johannesburg	1	9.6%	4.8%	15.9%	N/A	4.6%	1.6%	9.4%	0.3%
Gauteng	City of Tshwane(TSH)	1	18.3%	11.1%	27.2%	N/A	12.8%	6.5%	20.8%	0.9%
North West	Dr Kenneth Kaunda	1	13.4%	6.5%	23.0%	N/A	13.4%	5.8%	24.4%	0.9%
North West	Dr Ruth Segomotsi	1	11.2%	6.4%	17.5%	N/A	13.5%	7.5%	20.9%	0.9%
Western	Eden(DC4)	1	13.7%	6.8%	23.5%	N/A	9.8%	2.5%	25.0%	0.8%
Mpumalanga	Ehlanzeni(DC32)	1	10.7%	5.7%	17.3%	N/A	4.0%	1.5%	8.0%	0.2%
Gauteng	Ekurhuleni(EKU)	1	13.2%	6.5%	22.0%	N/A	5.5%	1.9%	11.3%	0.5%
Free State	Fezile Dabi(DC20)	1	12.7%	5.8%	23.0%	N/A	7.6%	2.0%	19.2%	0.7%
Northern	Frances Baard(DC9)	1	13.2%	7.4%	20.7%	N/A	5.6%	2.3%	10.6%	0.5%
Mpumalanga	Gert Sibande(DC30)	1	7.6%	3.8%	13.0%	N/A	4.1%	1.5%	8.2%	0.2%
Limpopo	Greater Sekhukhune	1	14.6%	8.1%	22.9%	N/A	7.9%	3.5%	14.3%	0.8%
Eastern	Joe Gqabi(DC14)	1	12.8%	6.8%	21.0%	N/A	4.6%	1.7%	9.6%	0.3%
Northern	John Taolo	1	8.4%	3.7%	15.3%	N/A	6.0%	2.1%	12.5%	0.5%
Free State	Lejweleputswa(DC18)	1	11.1%	5.0%	19.7%	N/A	7.0%	2.5%	14.6%	0.7%
Free State	Mangaung(MAN)	1	35.1%	21.0%	51.4%	N/A	7.6%	2.0%	19.6%	0.6%
Limpopo	Mopani(DC33)	1	8.2%	3.8%	14.5%	N/A	5.6%	2.1%	11.2%	0.5%
Northern	Namakwa(DC6)	1	14.6%	7.6%	24.1%	N/A	7.4%	2.8%	14.6%	0.7%
Eastern	Nelson Mandela	1	11.2%	5.6%	18.9%	N/A	5.5%	2.0%	11.2%	0.5%
North West	Ngaka Modiri	1	11.1%	5.8%	18.2%	N/A	5.7%	2.2%	11.0%	0.5%
Mpumalanga	Nkangala(DC31)	1	13.3%	7.3%	21.1%	N/A	11.7%	5.7%	20.0%	0.9%
Eastern	O.R.Tambo(DC15)	1	19.5%	12.6%	27.8%	N/A	3.5%	1.3%	7.0%	0.1%
Western	Overberg(DC3)	1	11.5%	5.8%	19.4%	N/A	5.4%	1.9%	10.9%	0.4%
Northern	Pixley ka Seme(DC7)	1	25.0%	13.7%	39.7%	N/A	8.3%	2.3%	20.3%	0.7%
Gauteng	Sedibeng(DC42)	1	16.5%	9.2%	26.2%	N/A	15.8%	7.9%	26.3%	0.9%
KwaZulu-	Sisonke(DC43)	1	18.4%	11.3%	26.9%	N/A	8.1%	3.8%	14.5%	0.8%
Northern	Siyanda(DC8)	1	13.3%	7.2%	21.2%	N/A	7.0%	2.7%	13.3%	0.7%
Free State	Thabo Mofutsanyane	1	12.8%	6.2%	22.1%	N/A	7.2%	2.0%	18.0%	0.7%
KwaZulu-	UMgungundlovu	1	8.5%	4.1%	14.5%	N/A	4.5%	1.6%	9.1%	0.3%

KwaZulu-	Ugu(DC21)	1	8.1%	4.4%	13.1%	N/A	2.9%	1.1%	5.8%	0.0%
KwaZulu-	Umkhanyakude(DC27	1	11.2%	5.6%	18.9%	N/A	8.3%	3.5%	15.9%	0.8%
KwaZulu-	Umzinyathi(DC24)	1	12.8%	4.9%	26.1%	N/A	5.5%	1.4%	14.4%	0.4%
KwaZulu-	Uthukela(DC23)	1	5.6%	3.0%	9.1%	N/A	4.2%	2.0%	7.5%	0.2%
KwaZulu-	Uthungulu(DC28)	1	11.4%	6.4%	17.8%	N/A	3.2%	1.2%	6.6%	0.1%
Limpopo	Vhembe(DC34)	1	25.5%	15.4%	38.0%	N/A	4.5%	1.5%	9.9%	0.3%
Limpopo	Waterberg(DC36)	1	12.0%	6.3%	19.6%	N/A	6.5%	2.7%	12.2%	0.6%
Western	West Coast(DC1)	1	10.4%	4.6%	18.7%	N/A	8.5%	2.3%	21.0%	0.7%
Gauteng	West Rand(DC48)	1	11.0%	5.0%	19.9%	N/A	8.9%	3.4%	18.0%	0.8%
Free State	Xhariep(DC16)	1	20.2%	11.3%	31.3%	N/A	7.1%	2.0%	17.3%	0.6%
KwaZulu-	Zululand(DC26)	1	8.2%	4.0%	14.0%	N/A	3.6%	1.3%	7.5%	0.1%
KwaZulu-	eThekwini(ETH)	1	8.1%	4.1%	13.7%	N/A	3.5%	1.3%	7.2%	0.1%
KwaZulu-	iLembe(DC29)	1	10.7%	5.7%	17.4%	N/A	5.1%	1.3%	13.2%	0.4%
Eastern	Alfred Nzo(DC44)	2	16.2%	8.8%	25.8%	N/A	7.3%	1.9%	18.1%	0.7%
KwaZulu-	Amajuba(DC25)	2	7.6%	4.0%	12.5%	N/A	6.6%	3.1%	11.6%	0.7%
Eastern	Amathole(DC12)	2	15.6%	5.8%	31.2%	N/A	30.8%	14.7%	51.8%	1.9%
North West	Bojanala(DC37)	2	10.8%	5.4%	18.4%	N/A	9.0%	2.4%	22.1%	0.7%
Eastern	Buffalo City(BUF)	2	16.2%	6.0%	33.4%	N/A	14.4%	3.0%	40.0%	0.8%
Eastern	Cacadu(DC10)	2	17.3%	9.9%	26.7%	N/A	9.8%	2.7%	24.0%	0.8%
Western	Cape Winelands(DC2	2	10.7%	5.5%	17.7%	N/A	16.9%	9.5%	26.4%	0.9%
Limpopo	Capricorn(DC35)	2	17.8%	10.2%	27.3%	N/A	8.5%	3.8%	15.5%	0.8%
Western	Central Karoo(DC5)	2	11.2%	5.7%	18.5%	N/A	10.1%	2.7%	24.5%	0.8%
Eastern	Chris Hani(DC13)	2	10.9%	5.7%	17.9%	N/A	9.0%	2.4%	22.0%	0.7%
Western	City of Cape Town	2	18.1%	11.2%	26.3%	N/A	14.7%	8.3%	22.7%	0.9%
Gauteng	City of Johannesburg	2	15.6%	9.5%	23.2%	N/A	9.9%	2.7%	24.2%	0.8%
Gauteng	City of Tshwane(TSH)	2	17.3%	10.7%	25.4%	N/A	9.2%	2.5%	22.2%	0.8%
North West	Dr Kenneth Kaunda	2	24.5%	13.9%	37.9%	N/A	10.1%	2.7%	24.2%	0.8%
North West	Dr Ruth Segomotsi	2	17.0%	10.5%	24.8%	N/A	12.4%	6.7%	19.7%	0.9%
Western	Eden(DC4)	2	22.8%	12.1%	36.7%	N/A	11.8%	3.1%	29.0%	0.8%
Mpumalanga	Ehlanzeni(DC32)	2	10.6%	5.8%	17.0%	N/A	6.0%	2.6%	11.1%	0.6%
Gauteng	Ekurhuleni(EKU)	2	10.8%	5.5%	18.2%	N/A	9.4%	2.5%	23.1%	0.8%
Free State	Fezile Dabi(DC20)	2	14.9%	5.7%	29.8%	N/A	9.4%	2.5%	23.3%	0.8%
Northern	Frances Baard(DC9)	2	14.2%	8.1%	21.9%	N/A	4.7%	1.8%	9.3%	0.3%
Mpumalanga	Gert Sibande(DC30)	2	14.2%	8.6%	21.0%	N/A	8.6%	2.3%	21.1%	0.7%
Limpopo	Greater	2	18.0%	11.2%	26.0%	N/A	5.8%	2.5%	10.8%	0.6%
Eastern	Joe Gqabi(DC14)	2	10.5%	5.3%	17.6%	N/A	7.6%	2.0%	19.1%	0.6%
Northern	John Taolo	2	9.1%	4.0%	16.7%	N/A	14.7%	6.6%	25.7%	0.9%
Free State	Lejweleputswa(DC18)	2	19.4%	11.1%	29.9%	N/A	13.0%	6.2%	22.4%	0.9%
Free State	Mangaung(MAN)	2	15.6%	7.4%	26.6%	N/A	9.4%	2.4%	23.5%	0.7%
Limpopo	Mopani(DC33)	2	12.2%	6.3%	19.9%	N/A	6.8%	2.7%	13.0%	0.7%
Northern	Namakwa(DC6)	2	14.7%	5.7%	29.1%	N/A	10.7%	3.0%	26.0%	0.8%
Eastern	Nelson Mandela	2	11.4%	6.0%	18.3%	N/A	6.0%	2.4%	11.6%	0.6%
North West	Ngaka Modiri	2	19.0%	11.9%	27.9%	N/A	11.7%	6.0%	19.3%	0.9%
Mpumalanga	Nkangala(DC31)	2	9.7%	4.8%	16.3%	N/A	9.3%	2.6%	22.1%	0.8%
Eastern	O.R.Tambo(DC15)	2	24.8%	16.8%	33.8%	N/A	4.4%	1.7%	8.6%	0.3%
Western	Overberg(DC3)	2	14.1%	5.2%	28.3%	N/A	8.8%	3.7%	16.3%	0.9%
Northern	Pixley ka Seme(DC7)	2	15.3%	5.9%	30.5%	N/A	10.1%	2.7%	24.1%	0.8%
Gauteng	Sedibeng(DC42)	2	14.7%	8.4%	22.9%	N/A	9.6%	4.5%	16.9%	0.9%

KwaZulu-	Sisonke(DC43)	2	20.0%	12.9%	28.6%	N/A	7.5%	1.9%	18.9%	0.6
Northern	Siyanda(DC8)	2	10.5%	5.2%	17.5%	N/A	26.1%	16.1%	38.3%	
Free State	Thabo	2	16.1%	8.2%	27.1%	N/A	13.7%	6.0%	25.5%	0.9
KwaZulu-	UMgungundlovu(DC22	2	19.4%	12.3%	28.3%	N/A	6.8%	1.8%	17.3%	0.6
KwaZulu-	Ugu(DC21)	2	17.2%	11.3%	23.9%	N/A	5.0%	2.3%	8.8%	0.4
KwaZulu-	Umkhanyakude(DC27	2	18.3%	9.6%	29.7%	N/A	7.1%	1.7%	18.6%	0.6
KwaZulu-	Umzinyathi(DC24)	2	14.5%	5.5%	29.2%	N/A	6.6%	1.7%	17.0%	0.5
KwaZulu-	Uthukela(DC23)	2	11.1%	7.3%	15.6%	N/A	3.0%	1.3%	5.5%	0.6
KwaZulu-	Uthungulu(DC28)	2	16.0%	9.5%	24.2%	N/A	6.0%	2.5%	11.2%	0.6
Limpopo	Vhembe(DC34)	2	31.6%	20.1%	44.8%	N/A	7.2%	1.8%	19.1%	0.6
Limpopo	Waterberg(DC36)	2	18.2%	11.1%	26.6%	N/A	7.1%	3.1%	12.8%	0.7
Western	West Coast(DC1)	2	13.8%	5.0%	28.6%	N/A	10.3%	2.6%	25.9%	0.8
Gauteng	West Rand(DC48)	2	11.1%	5.4%	19.3%	N/A	7.1%	2.7%	14.2%	0.7
Free State	Xhariep(DC16)	2	15.7%	6.0%	31.4%	N/A	8.7%	2.4%	21.7%	0.7
KwaZulu-	Zululand(DC26)	2	11.4%	6.0%	18.4%	N/A	6.1%	2.5%	11.9%	0.6
KwaZulu-	eThekwini(ETH)	2	15.5%	9.6%	22.6%	N/A	4.9%	2.1%	9.2%	0.4
KwaZulu-	iLembe(DC29)	2	8.9%	4.4%	14.9%	N/A	5.2%	2.1%	10.3%	0.4
Eastern	Alfred Nzo(DC44)	3	17.0%	9.7%	26.5%	N/A	6.2%	2.5%	12.3%	0.6
KwaZulu-	Amajuba(DC25)	3	12.2%	6.8%	19.4%	N/A	4.5%	1.8%	8.8%	0
Eastern	Amathole(DC12)	3	12.1%	5.4%	22.2%	N/A	10.1%	3.5%	20.9%	0
North West	Bojanala(DC37)	3	8.6%	4.1%	14.7%	N/A	5.7%	2.2%	11.1%	0.5
Eastern	Buffalo City(BUF)	3	14.3%	5.2%	30.4%	N/A	11.2%	2.2%	33.1%	0.7
Eastern	Cacadu(DC10)	3	9.8%	4.7%	16.9%	N/A	6.6%	2.6%	13.0%	0.7
Western	Cape Winelands(DC2	3	9.0%	4.3%	15.7%	N/A	5.0%	1.7%	10.3%	0.4
Limpopo	Capricorn(DC35)	3	17.1%	10.5%	25.3%	N/A	4.6%	1.8%	8.9%	0.3
Western	Central Karoo(DC5)	3	11.6%	6.0%	19.0%	N/A	4.9%	1.8%	10.0%	0.4
Eastern	Chris Hani(DC13)	3	12.4%	6.9%	19.6%	N/A	8.9%	4.2%	15.6%	0.9
Western	City of Cape	3	10.1%	5.7%	15.8%	N/A	4.8%	2.0%	9.0%	0
Gauteng	City of	3	14.7%	8.9%	21.8%	N/A	10.6%	5.6%	17.3%	0.9
Gauteng	City of Tshwane(TSH)	3	9.4%	5.2%	15.1%	N/A	6.1%	2.8%	10.9%	0
North West	Dr Kenneth	3	15.9%	8.0%	26.5%	N/A	7.6%	2.1%	19.0%	0.6
North West	Dr Ruth Segomotsi	3	21.0%	13.3%	30.0%	N/A	4.6%	1.7%	9.1%	0.3
Western	Eden(DC4)	3	10.6%	4.6%	19.4%	N/A	20.0%	9.4%	34.8%	0.9
Mpumalanga	Ehlanzeni(DC32)	3	26.6%	19.0%	35.2%	N/A	8.6%	4.5%	14.0%	0.9
Gauteng	Ekurhuleni(EKU)	3	8.0%	3.9%	13.7%	N/A	4.3%	1.6%	8.8%	0.3
Free State	Fezile Dabi(DC20)	3	13.4%	6.3%	23.9%	N/A	7.1%	1.9%	17.8%	0.6
Northern	Frances Baard(DC9)	3	9.1%	4.7%	15.3%	N/A	5.8%	2.4%	11.0%	0.6
Mpumalanga	Gert Sibande(DC30)	3	12.9%	7.6%	19.4%	N/A	13.3%	7.5%	20.6%	0.9
Limpopo	Greater	3	11.0%	6.6%	16.6%	N/A	4.1%	1.8%	7.6%	0.2
Eastern	Joe Gqabi(DC14)	3	18.8%	11.2%	28.5%	N/A	3.7%	1.3%	7.8%	
Northern	John Taolo	3	8.8%	4.1%	15.6%	N/A	5.9%	2.1%	11.9%	0.5
Free State	Lejweleputswa(DC18)	3	14.8%	7.9%	23.7%	N/A	7.4%	3.1%	13.8%	0
Free State	Mangaung(MAN)	3	17.6%	9.3%	28.5%	N/A	7.0%	1.7%	18.0%	0.6
Limpopo	Mopani(DC33)	3	23.9%	14.7%	34.6%	N/A	9.0%	3.9%	16.5%	
Northern	Namakwa(DC6)	3	13.3%	6.6%	22.3%	N/A	9.8%	4.1%	18.2%	0.9
Eastern	Nelson Mandela	3	16.1%	9.1%	25.0%	N/A	4.9%	1.7%	10.2%	0.4
North West	Ngaka Modiri	3	16.0%	9.8%	23.5%	N/A	6.6%	2.9%	11.8%	0.7
Mpumalanga	Nkangala(DC31)	3	7.3%	3.3%	13.2%	N/A	5.7%	2.1%	11.3%	0.5

Eastern	O.R.Tambo(DC15)	3	13.1%	7.4%	20.3%	N/A	3.2%	1.1%	6.5%	0.9%
Western	Overberg(DC3)	3	8.8%	4.3%	15.0%	N/A	7.3%	1.9%	18.6%	0.6%
Northern	Pixley ka Seme(DC7)	3	13.4%	5.0%	26.8%	N/A	15.5%	6.7%	28.7%	0.9%
Gauteng	Sedibeng(DC42)	3	10.6%	5.6%	17.2%	N/A	5.9%	2.4%	11.0%	0.7%
KwaZulu-	Sisonke(DC43)	3	14.6%	9.2%	21.3%	N/A	9.3%	5.0%	15.0%	0.9%
Northern	Siyanda(DC8)	3	14.1%	8.2%	21.8%	N/A	13.7%	7.5%	21.7%	0.9%
Free State	Thabo	3	17.3%	9.2%	28.2%	N/A	5.2%	1.7%	11.3%	0.4%
KwaZulu-	UMgungundlovu(DC22	3	12.9%	7.3%	20.2%	N/A	7.7%	3.5%	13.9%	0.8%
KwaZulu-	Ugu(DC21)	3	17.6%	11.8%	24.3%	N/A	5.3%	2.5%	9.3%	0.5%
KwaZulu-	Umkhanyakude(DC27	3	10.4%	5.6%	16.8%	N/A	3.2%	1.1%	6.7%	0.1%
KwaZulu-	Umzinyathi(DC24)	3	12.7%	4.9%	25.5%	N/A	5.0%	1.3%	12.9%	0.3%
KwaZulu-	Uthukela(DC23)	3	13.7%	9.5%	18.4%	N/A	3.4%	1.7%	5.9%	0.1%
KwaZulu-	Uthungulu(DC28)	3	16.1%	9.9%	23.6%	N/A	3.3%	1.2%	6.6%	0.1%
Limpopo	Vhembe(DC34)	3	7.5%	3.4%	13.7%	N/A	3.4%	1.1%	7.4%	0.1%
Limpopo	Waterberg(DC36)	3	12.5%	7.3%	19.5%	N/A	6.5%	1.8%	15.9%	0.5%
Western	West Coast(DC1)	3	7.5%	3.3%	13.6%	N/A	7.7%	3.1%	14.6%	0.8%
Gauteng	West Rand(DC48)	3	18.7%	10.2%	29.8%	N/A	4.9%	1.6%	10.4%	0.3%
Free State	Xhariep(DC16)	3	9.7%	4.5%	17.1%	N/A	4.8%	1.6%	10.1%	0.3%
KwaZulu-	Zululand(DC26)	3	12.8%	7.1%	20.0%	N/A	3.6%	1.3%	7.4%	0.1%
KwaZulu-	eThekwini(ETH)	3	9.5%	5.7%	14.5%	N/A	4.9%	2.3%	8.6%	0.4%
KwaZulu-	iLembe(DC29)	3	16.1%	9.0%	25.4%	N/A	4.4%	1.5%	9.4%	0.3%
Eastern	Alfred Nzo(DC44)	4	14.7%	8.2%	23.1%	N/A	2.8%	0.9%	6.3%	0.0%
KwaZulu-	Amajuba(DC25)	4	8.0%	4.5%	12.6%	N/A	3.9%	1.0%	10.1%	0.2%
Eastern	Amathole(DC12)	4	9.3%	3.9%	17.8%	N/A	6.9%	1.6%	17.7%	0.5%
North West	Bojanala(DC37)	4	8.9%	4.5%	15.0%	N/A	4.9%	1.3%	12.6%	0.3%
Eastern	Buffalo City(BUF)	4	10.0%	3.9%	19.9%	N/A	8.2%	1.6%	25.1%	0.6%
Eastern	Cacadu(DC10)	4	9.0%	4.4%	15.7%	N/A	4.2%	1.4%	8.7%	0.2%
Western	Cape Winelands(DC2	4	6.2%	3.0%	10.7%	N/A	6.2%	1.6%	15.6%	0.5%
Limpopo	Capricorn(DC35)	4	9.6%	5.1%	15.5%	N/A	3.4%	1.3%	6.9%	0.1%
Western	Central Karoo(DC5)	4	8.6%	4.1%	15.1%	N/A	4.6%	1.6%	9.4%	0.3%
Eastern	Chris Hani(DC13)	4	9.1%	4.7%	14.9%	N/A	3.8%	1.5%	7.6%	0.1%
Western	City of Cape	4	8.5%	3.0%	18.8%	N/A	7.0%	3.3%	12.2%	0.8%
Gauteng	City of	4	6.6%	3.5%	10.7%	N/A	8.4%	4.6%	13.4%	0.9%
Gauteng	City of Tshwane(TSH)	4	4.5%	2.1%	7.9%	N/A	3.7%	1.5%	7.0%	0.1%
North West	Dr Kenneth	4	6.9%	2.8%	13.1%	N/A	5.4%	1.8%	11.6%	0.4%
North West	Dr Ruth Segomotsi	4	8.0%	4.2%	13.3%	N/A	7.0%	3.3%	12.2%	0.8%
Western	Eden(DC4)	4	9.0%	3.3%	19.0%	N/A	6.6%	1.6%	17.3%	0.5%
Mpumalanga	Ehlanzeni(DC32)	4	6.6%	3.5%	10.8%	N/A	3.4%	1.4%	6.4%	0.1%
Gauteng	Ekurhuleni(EKU)	4	4.4%	2.0%	7.8%	N/A	6.5%	3.0%	11.5%	0.7%
Free State	Fezile Dabi(DC20)	4	6.7%	2.8%	12.8%	N/A	5.2%	1.3%	13.4%	0.4%
Northern	Frances Baard(DC9)	4	10.9%	6.1%	17.1%	N/A	3.4%	1.3%	7.0%	0.1%
Mpumalanga	Gert Sibande(DC30)	4	12.5%	7.3%	19.1%	N/A	4.7%	2.0%	9.0%	0.3%
Limpopo	Greater	4	11.9%	7.2%	17.6%	N/A	5.4%	2.5%	9.6%	0.5%
Eastern	Joe Gqabi(DC14)	4	16.4%	9.4%	25.3%	N/A	3.2%	1.1%	6.9%	0.1%
Northern	John Taolo	4	9.7%	5.2%	16.0%	N/A	8.4%	3.9%	14.6%	0.9%
Free State	Lejweleputswa(DC18)	4	13.2%	7.4%	20.8%	N/A	5.4%	2.2%	10.4%	0.5%
Free State	Mangaung(MAN)	4	8.5%	3.7%	15.8%	N/A	5.4%	1.8%	11.8%	0.4%
Limpopo	Mopani(DC33)	4	6.9%	3.3%	12.0%	N/A	5.3%	2.1%	10.3%	0.5%

Northern	Namakwa(DC6)	4	7.6%	3.4%	13.9%	N/A	7.1%	2.8%	13.8%	0.7
Eastern	Nelson Mandela	4	10.6%	5.5%	17.6%	N/A	4.2%	1.0%	11.9%	0
North West	Ngaka Modiri	4	9.4%	5.2%	15.0%	N/A	7.1%	3.4%	12.3%	0.8
Mpumalanga	Nkangala(DC31)	4	5.7%	2.6%	10.2%	N/A	5.1%	1.4%	12.8%	0.3
Eastern	O.R.Tambo(DC15)	4	8.6%	4.9%	13.3%	N/A	3.8%	0.9%	10.4%	0.2
Western	Overberg(DC3)	4	6.3%	2.9%	11.2%	N/A	3.1%	1.0%	6.6%	0
Northern	Pixley ka Seme(DC7)	4	7.6%	3.2%	14.2%	N/A	4.9%	1.7%	10.6%	0.4
Gauteng	Sedibeng(DC42)	4	8.7%	4.4%	14.6%	N/A	3.4%	1.2%	7.1%	0.1
KwaZulu-	Sisonke(DC43)	4	7.0%	3.7%	11.5%	N/A	2.5%	0.9%	5.2%	0.0
Northern	Siyanda(DC8)	4	9.0%	4.8%	14.9%	N/A	6.8%	3.0%	12.3%	0
Free State	Thabo	4	6.9%	3.0%	12.8%	N/A	4.8%	1.3%	12.3%	0.3
KwaZulu-	UMgungundlovu(DC22	4	6.4%	3.2%	10.9%	N/A	2.3%	0.8%	4.9%	0.0
KwaZulu-	Ugu(DC21)	4	7.3%	4.1%	11.4%	N/A	2.9%	1.2%	5.6%	0.0
KwaZulu-	Umkhanyakude(DC27	4	7.4%	3.6%	12.4%	N/A	3.8%	1.0%	10.2%	
KwaZulu-	Umzinyathi(DC24)	4	8.8%	3.3%	18.9%	N/A	3.6%	0.9%	9.8%	0.1
KwaZulu-	Uthukela(DC23)	4	9.6%	6.4%	13.3%	N/A	5.0%	2.8%	7.8%	0.4
KwaZulu-	Uthungulu(DC28)	4	11.4%	6.9%	17.2%	N/A	3.2%	1.3%	6.4%	0
Limpopo	Vhembe(DC34)	4	13.5%	7.8%	20.7%	N/A	2.4%	0.7%	5.2%	0.0
Limpopo	Waterberg(DC36)	4	8.2%	4.4%	13.1%	N/A	3.5%	1.4%	6.7%	0.1
Western	West Coast(DC1)	4	5.2%	2.2%	9.9%	N/A	4.5%	1.6%	9.5%	
Gauteng	West Rand(DC48)	4	6.2%	2.6%	11.9%	N/A	4.1%	1.3%	8.9%	0.2
Free State	Xhariep(DC16)	4	13.4%	6.9%	22.1%	N/A	4.8%	1.3%	12.4%	0.3
KwaZulu-	Zululand(DC26)	4	8.3%	4.5%	13.2%	N/A	2.9%	1.1%	5.9%	0.0
KwaZulu-	eThekwini(ETH)	4	12.7%	8.2%	17.9%	N/A	2.0%	0.8%	4.1%	0.0
KwaZulu-	iLembe(DC29)	4	7.0%	3.6%	11.6%	N/A	2.8%	1.0%	5.7%	0.0
Eastern	Alfred Nzo(DC44)	5	6.6%	2.7%	12.7%	0.958	3.6%	1.1%	8.1%	0.1
KwaZulu-	Amajuba(DC25)	5	8.9%	4.4%	15.0%	0.6443	3.1%	1.1%	6.7%	0.1
Eastern	Amathole(DC12)	5	8.2%	3.1%	17.0%	0.669	6.1%	1.6%	15.4%	0.5
North West	Bojanala(DC37)	5	8.7%	3.9%	16.4%	0.3584	4.6%	1.2%	12.2%	0.3
Eastern	Buffalo City(BUF)	5	9.3%	3.2%	20.6%	0.6684	7.9%	1.4%	25.4%	0
Eastern	Cacadu(DC10)	5	8.1%	3.5%	15.3%	0.5055	4.9%	1.3%	12.7%	0.3
Western	Cape Winelands(DC2	5	9.6%	4.4%	17.0%	0.3039	5.8%	2.2%	11.7%	0.5
Limpopo	Capricorn(DC35)	5	11.9%	5.9%	20.2%	0.6993	3.7%	1.2%	8.1%	0.1
Western	Central Karoo(DC5)	5	7.5%	3.2%	14.4%	0.7188	6.4%	2.3%	13.8%	0.6
Eastern	Chris Hani(DC13)	5	6.2%	2.6%	11.8%	0.8805	3.5%	1.1%	7.7%	0.1
Western	City of Cape	5	8.3%	3.8%	14.8%	0.5419	6.8%	2.7%	13.4%	0.7
Gauteng	City of	5	7.6%	3.7%	13.3%	0.8987	4.3%	1.5%	8.7%	0.2
Gauteng	City of Tshwane(TSH)	5	6.6%	3.2%	11.5%	0.6727	3.5%	1.2%	7.1%	0.1
North West	Dr Kenneth	5	11.8%	5.3%	21.7%	0.6061	5.0%	1.5%	11.5%	0.4
North West	Dr Ruth Segomotsi	5	7.2%	3.2%	13.2%	0.9883	6.0%	2.3%	11.9%	0.5
Western	Eden(DC4)	5	6.8%	2.7%	13.4%	0.7027	6.1%	1.4%	15.7%	0.5
Mpumalanga	Ehlanzeni(DC32)	5	7.2%	3.4%	12.7%	0.9998	3.4%	1.2%	7.1%	0.1
Gauteng	Ekurhuleni(EKU)	5	5.6%	2.4%	10.5%	0.6546	5.9%	2.3%	11.7%	0.6
Free State	Fezile Dabi(DC20)	5	8.9%	3.7%	17.5%	0.6807	4.8%	1.3%	12.4%	0.3
Northern	Frances Baard(DC9)	5	9.5%	4.6%	16.6%	0.3076	4.0%	1.4%	8.6%	0.2
Mpumalanga	Gert Sibande(DC30)	5	9.0%	4.3%	15.9%	0.6795	3.4%	1.1%	7.5%	0.1
Limpopo	Greater	5	6.8%	3.0%	12.3%	0.7849	4.7%	1.7%	9.8%	0.3
Eastern	Joe Gqabi(DC14)	5	7.4%	3.0%	14.0%	0.9632	3.2%	0.9%	7.5%	0.1

Northern	John Taolo	5	7.1%	3.0%	13.7%	0.5346	5.5%	1.8%	11.8%	0.4
Free State	Lejweleputswa(DC18)	5	11.7%	5.7%	20.3%	0.5537	4.1%	1.3%	9.0%	0.2
Free State	Mangaung(MAN)	5	9.8%	4.1%	18.4%	0.8263	4.7%	1.2%	12.4%	0.3
Limpopo	Mopani(DC33)	5	10.0%	4.6%	17.9%	0.9661	5.2%	1.8%	11.2%	0.4
Northern	Namakwa(DC6)	5	8.2%	3.0%	17.9%	0.731	4.9%	1.5%	11.2%	0.4
Eastern	Nelson Mandela	5	8.9%	4.0%	16.1%	0.8491	3.2%	0.9%	7.5%	0
North West	Ngaka Modiri	5	7.0%	3.1%	12.9%	0.95	4.4%	1.5%	9.1%	0.3
Mpumalanga	Nkangala(DC31)	5	8.9%	4.0%	16.6%	0.2224	5.1%	1.8%	11.0%	0.4
Eastern	O.R.Tambo(DC15)	5	10.1%	5.4%	16.4%	0.5924	2.3%	0.7%	5.1%	0.0
Western	Overberg(DC3)	5	9.0%	4.1%	16.1%	0.3381	4.9%	1.2%	12.9%	0.3
Northern	Pixley ka Seme(DC7)	5	9.7%	4.0%	18.7%	0.5992	5.2%	1.4%	13.3%	0.4
Gauteng	Sedibeng(DC42)	5	6.4%	2.6%	12.1%	0.7712	6.6%	2.4%	13.3%	0.6
KwaZulu-	Sisonke(DC43)	5	7.7%	3.5%	14.2%	0.887	2.8%	0.8%	6.3%	0.0
Northern	Siyanda(DC8)	5	11.7%	5.9%	19.6%	0.5127	5.2%	1.9%	10.6%	0.4
Free State	Thabo	5	9.0%	3.7%	17.3%	0.8498	4.4%	1.2%	11.4%	0.3
KwaZulu-	UMgungundlovu(DC22	5	6.6%	2.9%	12.2%	0.8795	3.8%	1.3%	8.2%	0.2
KwaZulu-	Ugu(DC21)	5	7.6%	3.8%	12.9%	0.9731	3.3%	1.2%	6.7%	0.1
KwaZulu-	Umkhanyakude(DC27	5	7.8%	3.6%	14.2%	0.6034	4.4%	1.5%	9.7%	0.3
KwaZulu-	Umzinyathi(DC24)	5	6.4%	2.9%	11.5%	0.8143	3.4%	1.2%	7.3%	0.1
KwaZulu-	Uthukela(DC23)	5	7.2%	3.5%	12.5%	0.9174	3.2%	1.1%	6.8%	0.1
KwaZulu-	Uthungulu(DC28)	5	11.1%	5.8%	18.0%	0.7098	3.0%	1.0%	6.5%	0.0
Limpopo	Vhembe(DC34)	5	9.7%	4.5%	17.3%	0.1707	4.3%	1.4%	9.5%	0.3
Limpopo	Waterberg(DC36)	5	6.0%	2.7%	10.9%	0.913	4.1%	1.5%	8.4%	0.2
Western	West Coast(DC1)	5	6.6%	2.6%	13.2%	0.4791	4.5%	1.3%	10.6%	0
Gauteng	West Rand(DC48)	5	6.3%	2.5%	12.9%	0.9742	3.7%	1.1%	8.6%	0.2
Free State	Xhariep(DC16)	5	8.5%	3.6%	16.2%	0.4699	4.4%	1.2%	11.5%	0.3
KwaZulu-	Zululand(DC26)	5	7.6%	3.6%	13.5%	0.7981	2.3%	0.7%	5.2%	0
KwaZulu-	eThekwini(ETH)	5	6.7%	3.2%	11.5%	0.6792	2.4%	0.8%	5.2%	0.0
KwaZulu-	iLembe(DC29)	5	7.0%	3.2%	12.9%	0.9372	3.2%	0.8%	8.7%	0.1

Reviewer: 2

The authors have adequately addressed my previous comments.

Response: Many thanks.

VERSION 3 – REVIEW

REVIEWER	Brandon Parkes Imperial College London, UK
REVIEW RETURNED	16-Mar-2020
GENERAL COMMENTS	I believe my previous comments have now been addressed sufficiently.